# Cortical recruitment determines learning dynamics and strategy

Sebastian Ceballo[1], Jacques Bourg [1], Alexandre Kempf[1], Zuzanna Piwkowska [1,3], Aurélie Daret[1], Pierre Pinson[1], Thomas Deneux[1], Simon Rumpel [2] & Brice Bathellier [1]

Salience is a broad and widely used concept in neuroscience whose neuronal correlates, however, remain elusive. In behavioral conditioning, salience is used to explain various effects, such as stimulus overshadowing, and refers to how fast and strongly a stimulus can be associated with a conditioned event. Here, we identify sounds of equal intensity and perceptual detectability, which due to their spectro-temporal content recruit different levels of population activity in mouse auditory cortex. When using these sounds as cues in a Go/ NoGo discrimination task, the degree of cortical recruitment matches the salience parameter of a reinforcement learning model used to analyze learning speed. We test an essential prediction of this model by training mice to discriminate light-sculpted optogenetic activity patterns in auditory cortex, and verify that cortical recruitment causally determines association or overshadowing of the stimulus components. This demonstrates that cortical recruitment underlies major aspects of stimulus salience during reinforcement learning.

[1] Paris-Saclay Institute of Neuroscience (NeuroPSI), Department for Integrative and Computational Neuroscience (ICN), UMR9197 CNRS/University Paris Sud, CNRS Bldg. 32/33, 1 Av. de la Terrasse, 91190 Gif-sur-Yvette, France. [2] Institute of Physiology, Focus Program Translational Neuroscience, University Medical Center, Johannes Gutenberg University, D-55099 Mainz, Germany. [3] Present address: Institut Pasteur, Dynamic Neuronal Imaging Unit, Paris 75015, France. These authors contributed equally: Sebastian Ceballo, Jacques Bourg, Alexandre Kempf. Correspondence and requests for materials should be addressed to B.B. (email: brice.bathellier@unic.cnrs-gif.fr)

Sensory stimuli can vary in their efficacy as a conditioned stimulus during behavioral conditioning. In classical conditioning, a well-known example is the so called "overshadowing" effect. When animals are trained to associate two simultaneously presented stimuli (historically a tone and a flash) to a specific unconditioned stimulus (e.g. foot-shock), it is often observed that, after training, the animal is conditioned more strongly to one stimulus than to the other[1,2]. In their theoretical work originally developed for classical conditioning, but later extended to operant conditioning, Rescorla and Wagner[3] introduced the notion of salience to explain the overshadowing phenomenon. In their model, salience is a parameter affecting the speed at which a given stimulus is associated with the unconditioned stimulus. Thus, when behavior reaches maximal performance and learning stops, the more salient of the two stimuli has been associated more strongly with the unconditioned stimulus, leading to overshadowing. While this theory captures a number of phenomena and is the basis for important frameworks such as reinforcement learning[4,5], the neural underpinnings of the salience parameter remain elusive.

Salience in this context is usually seen as the global amount of neural activity representing the stimulus, like in models of attentional salience[6–9]. This intuitively follows from the idea that if more spikes represent a stimulus, they can produce more synaptic weight changes, as expected from the firing rate sensitivity of typical learning rules[10–14], and thus modulate more rapidly the relevant connections. However widespread, this idea lacks direct causal experimental verification in a learning task. Moreover, other theories propose that salience could also be encoded in other parameters such as neuronal synchrony levels[15–18], which could influence learning via the temporal properties of biological learning rules[19–23]. Thus, the neuronal correlate of stimulus salience is a key question with broad implications for learning theories.

Using auditory discrimination tasks of sounds with different global cortical response strengths, we show that cortical recruitment impacts learning dynamics[24,25] similarly to the salience parameter of a reinforcement learning model. To explore this result in more precise experimental settings, we trained mice to discriminate optogenetically driven response patterns that elicit different levels of cortical activity. Using this paradigm, we directly demonstrate that cortical recruitment determines which part of a compound stimulus drives a learned association while "overshadowing" other parts of the stimulus. This validates a generic prediction of reinforcement learning models and causally establishes the role of cortical recruitment as a neuronal correlate of stimulus salience.

## Results

**Sounds with identical levels can recruit different activity levels.** To investigate the relationship between stimulus salience and neuronal recruitment, we first aimed to identify sounds recruiting different amounts of cortical activity. A previous report has shown that complex sounds with different frequency content but equal duration and sound pressure level can recruit population responses of different sizes in cat auditory cortex[26]. To test if a similar phenomenon exists in mice, which would then allow us to experimentally decouple recruitment from physical intensity, we chose three short, complex sounds (70 ms duration) containing a large range of frequencies and temporal modulations, but normalized at equal mean pressure level (73 dB SPL, Fig. 1a). These sounds displayed different power spectra in the 10–30 kHz range (Fig. 1a) where the mouse ear is most sensitive[27–29]. We thus wondered if this discrepancy was affecting their detectability. To do so, we trained mice to lick on a water port after presentation of any of the three sounds randomly presented in the same task to obtain a reward (Fig. 1b). We then measured response probability to decreasing intensity levels. All mice experience the three sounds in the same task. We observed that, for all sounds, response probability steadily decreased down to chance level as measured in the absence of sound (Fig. 1c). Yet, no significant response difference was observed across the three sounds (Fig. 1c), indicating that the chosen 73 dB SPL was at a comparable distance from the detection threshold for the three sounds.

We assessed recruitment of neural activity in the auditory cortex in response to these three sounds using two-photon calcium imaging in awake, passively listening mice. We imaged six mice that were injected with AAV1-GCAMP6s virus in auditory cortex (Fig. 2a). Recordings were followed by an automated image registration and segmentation algorithm (Fig. 2b)[30] that allowed the isolation of 15,511 neurons across 27 imaging sites, from which large fluorescence signals could be observed (Fig. 2c). The fields-of-view were either $0.5 \times 0.5$ or $1 \times 1$ mm (Fig. 2a, b), allowing a rapid tiling of the full extent of primary and secondary auditory cortex (Supplementary Fig. 1). Cortical depths were randomly chosen ranging between 100 and 300 μm corresponding to layer 2/3. The mouse auditory cortex (primary+secondary) contains ~200,000 neurons in one hemisphere[31] and thus ~50,000 neurons in layer 2/3, so we expect our sample of ~15,000 neurons to be representative for supragranular auditory cortex. As typical learning rules are dependent on presynaptic firing rate[10–14], we first measured the amplitude of the mean-deconvolved calcium signals, a proxy for neuronal firing rate[32], recorded across the entire duration of the response (Fig. 2d). We observed that at 73 dB intensity, sound A elicited at least twofold less cortical activity than sounds B and C (Fig. 2e).

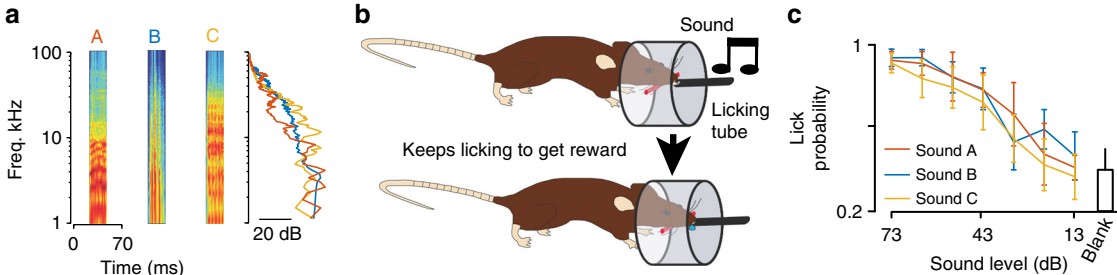

**Fig. 1** Spectro-temporal differences in complex sounds do not affect near-threshold detectability. **a** Spectrograms of three 70-ms-long complex sounds, with power spectrum on the right. **b** Schematics describing the auditory detection task. **c** Mean response probability for six mice trained to detect sounds A, B, and C at 73 dB to get a reward and probed with lower sound intensities. While the effect of intensity was significant, there was no effect of sound identity (Friedman test, $p_{intensity} = 2.3 \times 10^{-9}$, $p_{sound} = 0.43$, $n = 6$ mice). Error bars represent standard errors (SEM). Source data are provided as a Source Data file

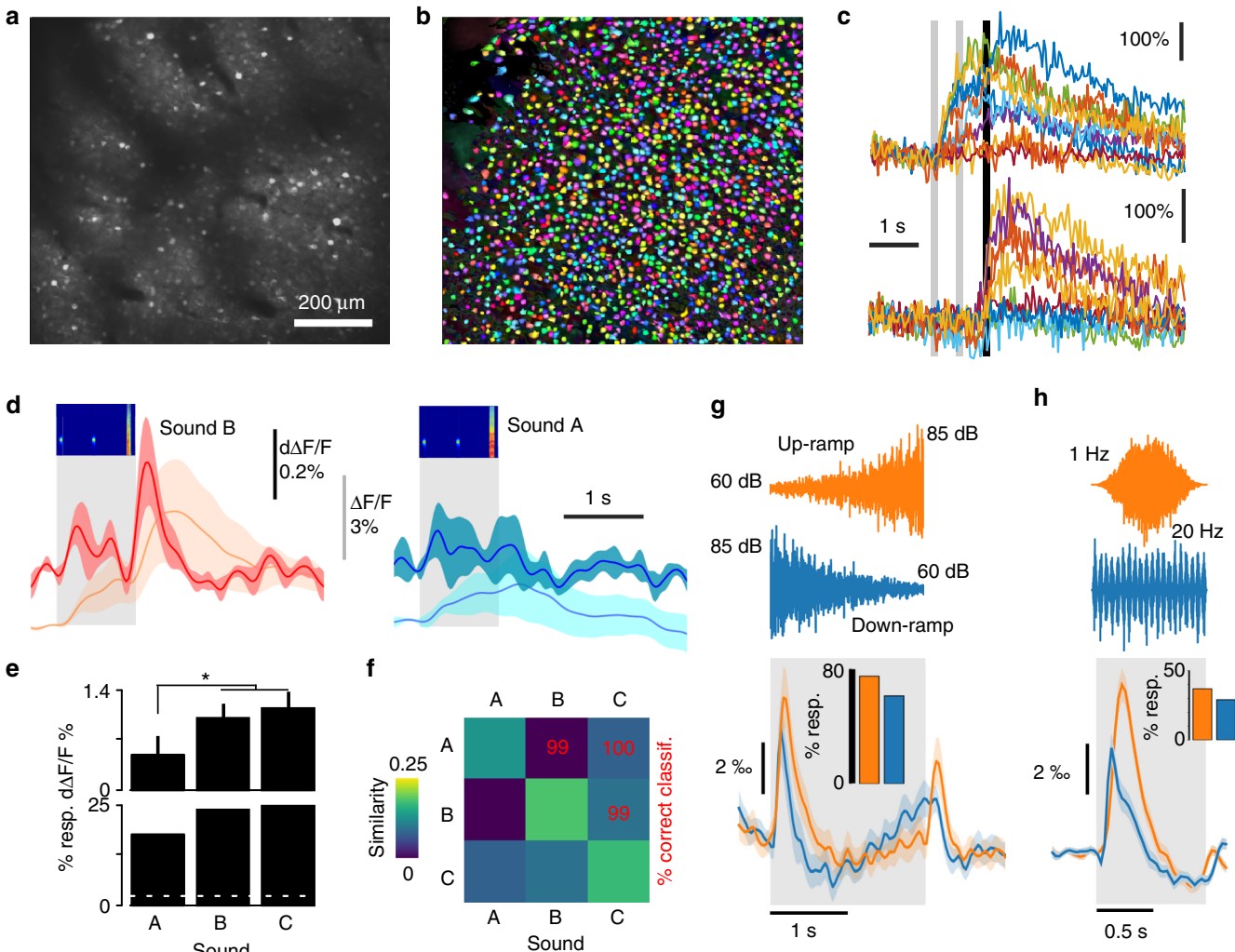

**Fig. 2** Spectro-temporal differences impact on cortical recruitment. **a** Example field of view illustrating GCAMP6s labeling of L2/3 auditory cortex neurons. **b** Result of automated cell segmentation run on the data acquired in the example shown in **a**. **c** Example single trial responses to sounds (different colors) for two neurons (top and bottom). Gray bars = sound duration. **d** Population responses (n = 27 sessions, 15,511 neurons in 6 mice) to Sound B (red) and A (blue). Both normalized fluorescence (light colors) and deconvolved (dark colors) calcium signals are shown. **e** Mean deconvolved signal and fraction of significantly responding neurons to sounds A, B, and C. Mean calcium responses to sound A (0.05 ± 0.03% ΔF/F.s$^{-1}$, mean±SEM) were significantly smaller than to B (0.10 ± 0.02% ΔF/F.s$^{-1}$) and C (0.12 ± 0.02% ΔF/F.s$^{-1}$; sign test, p = 0.0008 and p = 0.026, n = 27 sessions, 15,511 neurons in 6 mice). Sound B and C also activated a larger fraction of neurons (24 and 25%; two-sided Signed test across 20 sound repetitions, p < 0.05) than A (18%; $\chi^2$ test, p = 10$^{-41}$ and 6 × 10$^{-31}$, n = 15,511). **f** Population response reliability (diagonal) and similarity (off-diagonal) matrix for sounds A, B, and C. The pair-wise discriminability value, computed with a linear classifier is indicated in red. **g** Mean deconvolved calcium signals for 6757 auditory cortex neurons in 12 awake mice during 29 calcium imaging sessions for 2-s-long white noise sounds modulated in intensity between 60 dB and 85 dB upwards and downwards (mean between 0 and 0.5 s after sound onset for up: 0.80 ± 0.08% vs down: 0.62 ± 0.11%, Signed test p = 3.75 × 10$^{-4}$, n = 29 sites, in 12 mice; mean between 0 and 2.5 s: 0.166 ± 0.58% vs down: 0.157 ± 0.57%, Signed test, p = 0.034). The inset show the significantly different ($\chi^2$ test, p = 10$^{-68}$) fraction of responsive cells (75 and 61%; two-sided Signed test, p < 0.05). **h** Mean deconvolved calcium signals for 59,590 auditory cortex neurons in seven awake mice during 60 calcium imaging sessions for white noise sounds modulated in intensity at 1Hz and at 20Hz (0.357 + −0.032% and 0.15 + − 0.034% ΔF/F.s$^{-1}$, signed test, p = 0.0009). The inset show the significantly different ($\chi^2$ test, p = 10$^{-189}$) fraction of responsive cells (37 and 29%; two-sided Signed test, p < 0.05). Error bars represent standard errors (SEM)

This was consistently observed across mice (Supplementary Fig. 1). Sounds producing more firing in the population activated also more neurons (~18% for sound A and ~25% for B and C, Fig. 2e). But note that the fraction of responsive neurons strongly depends on statistical threshold. Furthermore, the observed differences in neuronal activity recruitment was consistent with previous, independent measurements performed under anesthesia (Supplementary Fig. 1)[33].

All three sounds elicited distinct response patterns as evaluated by correlation-based population similarity measures and sound identity could be decoded with high accuracy based on single-trial

response patterns using linear classifiers (Fig. 2f), indicating that sound discriminability was not affected by cortical recruitment.

Another discrepancy between cortical recruitment and the physical intensity of a stimulus can be observed using sounds with different temporal intensity profiles. Up-ramping sounds elicit larger cortical responses in mice[34] and other animals[35,36] than their time-symmetric down–ramps, despite equal physical energies. This effect correlates with asymmetries in subjectively perceived loudness in humans[37,38]. We confirmed this result for 2 s white noise sounds ramping between 60 and 85 dB, with a clear effect at sound onset despite the lower initial intensity level

in up-ramps (Fig. 2g). Rhythmic amplitude modulations provided another striking example, as we observed that a white noise sound modulated at 1λ produces more activity than when modulated at 20Hz, although the two sounds have the same physical energy (Fig. 2h).

In summary, when different sounds are played at an intensity above the detection threshold, the amount of recruited cortical activity in mouse auditory cortex depends on factors other than intensity and can vary across different sounds. Based on this observation, we asked whether cortical recruitment could be related to stimulus salience in a learning task.

**Cortical recruitment influences learning speed**. Classically, relative salience measures are performed using an overshadowing paradigm in which two stimuli are conditioned together, as a compound stimulus, to an unconditioned stimulus. Then, salience is derived from the level of the conditioned response elicited by each stimulus component individually. While this approach is valid when the compound is made of stimuli from two different sensory modalities, two simultaneous sounds are likely to fuse perceptually, precluding measurement of their individual saliences with the classical overshadowing design[39]. Alternatively, Rescorla and Wagner's model[3] postulates that learning speed follows stimulus salience. We thus decided to test if learning speed relates to cortical recruitment, using an auditory-cued Go/No-Go task. To do so, water-deprived mice were first trained to visit a lick-port and to receive a water reward if they licked after being presented with an S+ sound. The main effect of this pre-training phase was to raise motivation rather than to learn sound-reward association and thus could not be used to measure learning speed. When mice collected rewards in at least 80% of their port visits, the Go/NoGo task was started by introducing a non-rewarded S− sound in half of the trials (Fig. 3a). After a large number of trials, mice succeeded to both sustain licking to the S+ and withdraw from licking for the S− (Fig. 3b), demonstrating discrimination of the two sounds. Importantly, as typically observed in such tasks[25], the S+ sound was rapidly associated with the lick response and the rate limiting factor in task learning was to associate the suppression of licking with the S− sound (Fig. 3b). Hence, learning speed depends more on the properties of the S− than of the S+ sound in this task. Discrepancies between two stimuli X and Y can thus be measured by comparing the learning speed of the X versus Y Go/NoGo discrimination when X is the S− against the speed observed when Y is the S−. For example, if X is less salient than Y, we expect learning to be slower when X is the S−. We therefore trained eight cohorts of mice to compare learning speed for sounds pairs A-B, A-C, B-C, and for the pair of sinusoidaly modulated sounds. We also used learning speed data from an earlier study for discrimination of up and down-ramping sounds[34]. Plotting the population learning curves for the sound pairs with maximum cortical recruitment differences (A–B and A–C,), we qualitatively observed that the average learning speed was faster when cortical recruitment for the S− sound was larger than for the S+ sound (Fig. 3c), suggesting a link between learning speed and cortical recruitment.

However, looking at learning curves from individual mice, we noticed that the qualitative difference observed at the group level hides a more complex effect. As often observed in animal training[24] and as we previously reported for our task[25], most individual learning curves had a sigmoidal rather than exponential time course. Specifically, the curves displayed a delay phase with no increase in performance followed by a learning phase with a steep performance increase. Also, the duration of each phase was highly variable across animals as exemplified in Fig. 3b. We wondered whether cortical recruitment was affecting one

particular phase or both. Using sigmoidal fits (Fig. 3b), we measured the delay phase duration as the number of trials necessary to reach 20% of maximal performance, and the learning phase duration as the number of trials necessary to go from 20 to 80% maximal performance. We observed across the five sound pairs tested that learning phase duration was systematically longer when the S− sound recruited less activity than the S+ sound (Fig. 3d) and a non-parametric analysis of variance showed this effect to be highly significant across all mouse populations tested. No systematic effect of cortical recruitment was observed for the delay phase (Fig. 3d, Supplementary Fig. 2). In addition, we noticed that cortical recruitment had also an effect on inter-individual variability. When the S− sound recruited more activity than the S+ sound, learning phase duration was more homogenous than for the opposite sound assignment, especially for the two sound pairs with a large difference of cortical recruitment (mean normalized standard deviation difference: 93% ± 18%, mean±SEM, $n = 5$ sound pairs, $p = 0.008$ Wilcoxon rank-sum test, Fig. 3d, e and Supplementary Fig. 2). Together, these results obtained over a total of 72 mice, indicated clear relationships between cortical recruitment and learning phase duration for the five pairs of sounds tested. It cannot be ruled out a priori that other non-measured parameters of the sound representations could explain this dependency. Yet, in the hypothesis that these parameters would be randomly assigned to the tested sounds, the probability to obtain by chance a consistent result across five independent experiments would be only about 3% ($2^{-5}$), so we expect this eventuality to be rather unlikely.

**A reinforcement learning model reproduces recruitment effects**. To theoretically evaluate the generality of these results, we tested a recent model of the discrimination task, extending the Rescorla–Wagner reinforcement learning framework to a simple but more biologically interpretable neuronal model (Fig. 4a)[25]. The model postulates that associative learning occurs by adjusting the synaptic weights between sensory and decision neural populations described by population firing rate variables. At the input, two populations are specific for the S+ and S− sounds respectively, which we denote as Ŝ+ and Ŝ−, and one population, Ĉ, which represents information common to S+ and S− trials (e.g. overlap between the S+ and S− representations or activity independent of sound, for example, related to visiting the lick port). Population Ĉ is an essential element of the model to reproduce high initial hit-rates during delayed discrimination learning[25]. The decision population has two ensembles: one promoting and one inhibiting licking (Fig. 4a). Adjustment of synaptic weights happens through a Hebbian learning rule modulated by Rescorla and Wagner's (1972) δ-rule which gates weight updates by the reward expectation error. However, the employed δ-rule is asymmetric, meaning that the learning rate is larger by a factor $v$ when an unexpected reward occurs, as compared to when an expected reward does not occur. This asymmetry is crucial for capturing the fast raise of the hit-rate, and slower adjustment of the correct rejection-rate (Fig. 3b). Last, synaptic updates are multiplicative, meaning that weight updates are proportional to the current weight value[40–43]. The key feature of multiplicative learning is that learning speed depends on the current strength of the synapses. Thus, the same model can have very slow learning (low weights) as in the delay phase and faster learning (high weights) as in the learning phase. Furthermore, this feature makes learning dynamics highly sensitive to the initial synaptic weights, which become important parameters that can even account for most of the inter-individual variability[25]. Importantly, the activity level of the Ŝ+ and Ŝ−

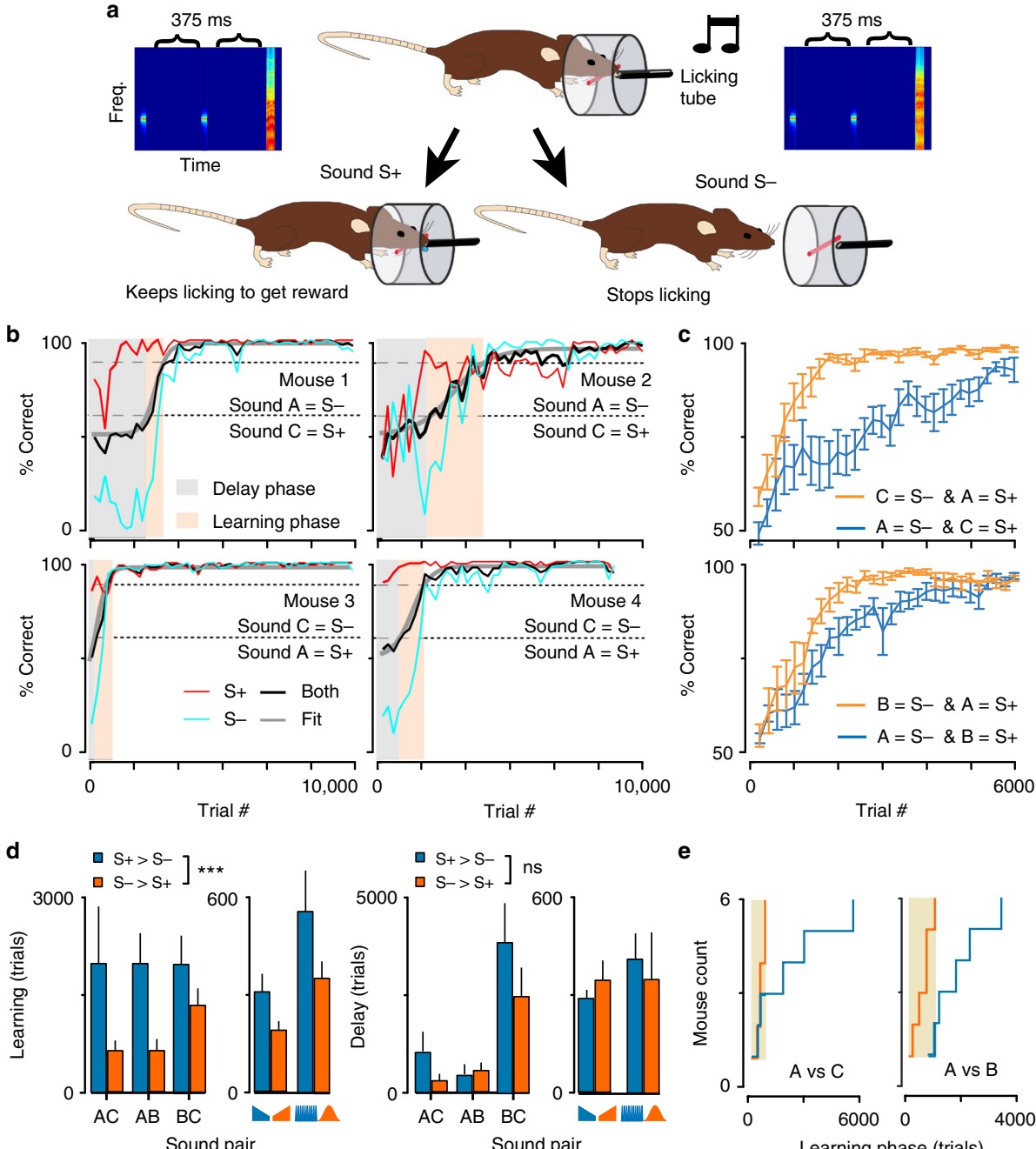

**Fig. 3** Cortical recruitment differences impacts learning phase duration. **a** Schematics describing the auditory Go/NoGo discrimination task. **b** Individual learning curves for four mice discriminating sounds A and C. Performance for S+ (red), S− (light blue) and both (black) sounds are displayed. Mice from the top row have sound A as the S− stimulus while mice from the bottom row have sound C as the S− stimulus. Typical learning curves display a delay and learning phase as shown in light gray and orange colors. **c** Mean learning curves for different groups of mice (n = 6 for each curve) discriminating between sounds A and C (top) or A and B (bottom). Slower learning is observed when the S− sound recruits less cortical activity than the S+ sound (blue) as compared to when sound valence is swapped (orange). **d** Mean±standard error for the learning and delay phase for the five discriminated sound pairs (A vs C, A vs B, B vs C, up- vs down ramp, 20Hz vs 1Hz modulation represented by blue and orange symbols). The conditions "S− recruitment >S+ recruitment" (blue) and "S+ recruitment <S− recruitment" (orange) are significantly different for the learning phase but not the delay phase (Friedman test, p = 0.0005 indicated as *** and p = 0.72 indicated as ns, n = 6 mice per group except for the up- and down ramps, n = 12). **e** Cumulative distributions of learning phase durations for sound pairs A-C and A-B. Error bars represent standard errors (SEM)

populations can be varied in the model, allowing to simulate the impact of cortical recruitment, for example, by setting either Ŝ− or Ŝ+ activity level to 2 while level for the other population is 1 (but note that this value can be widely varied without changing our conclusions, Fig. 4b, Supplementary Figs. 3 and 4, Supplementary Note 1).

We therefore wondered whether this model reproduces the observed relationships between cortical recruitment and mean learning speed and its variability. The dynamics of the model depend on the choice of its three core parameters (noise level, learning rate, and asymmetry, see Methods) and of its initial synaptic weights. We showed previously[25] that individual learning

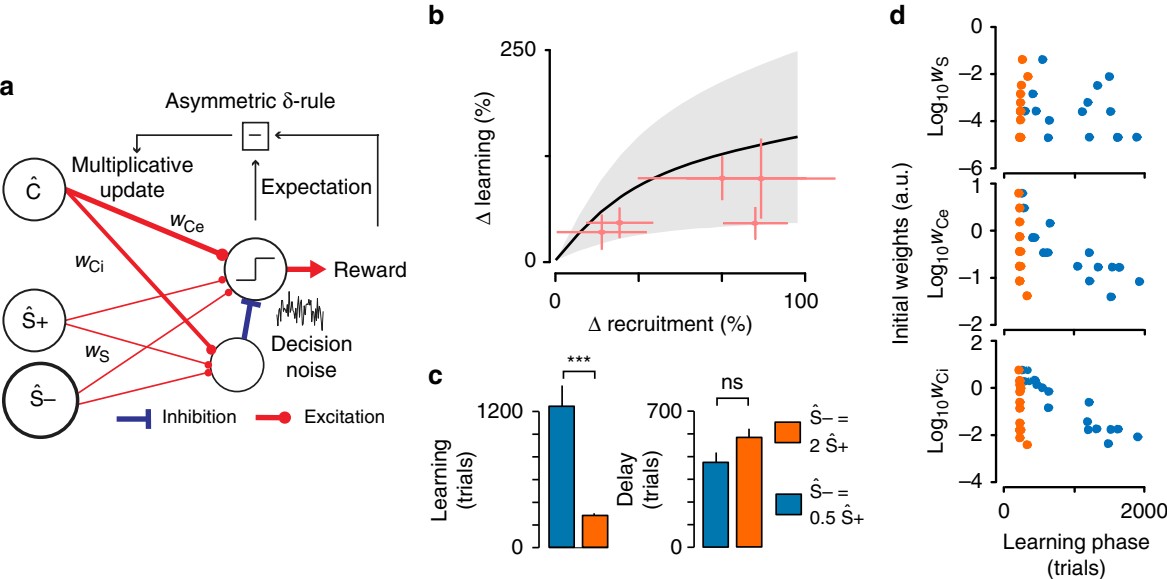

**Fig. 4** A multiplicative reinforcement learning model reproduces modulations of learning speed by neuronal recruitment. **a** Schematics describing the auditory Go/NoGo discrimination model. **b** Mean±standard deviation of the difference of learning phase duration against difference in neural recruitment by the stimuli for the model initialized with 15 sets of initial conditions obtained when fitting learning curves from a previous study[25]. The experimental observations of Fig. 3d are superimposed in red. **c** Mean±standard errors for the learning and delay phases obtained with the model for a twofold difference in cortical recruitment between the two stimuli. A longer learning phase (Kolmogorov–Smirnov test, $p = 8 \times 10^{-7}$ indicated as ***, $n = 15$ initial conditions per group) but not delay phase (Kolmogorov–Smirnov test, $p = 0.13$ indicated as ns, $n = 15$ initial conditions per group) is observed when S− recruits less activity (blue) as compared to when S− recruits more activity than S+ (orange). **d** Learning phase duration plotted against the value of the modeled initial synaptic weights. Same simulations and color code as in **c**. Significant correlations were observed only when S− recruits less activity (blue), and for $w_{Ce}$ ($\rho = -0.61$, $p = 0.015$, $n = 15$) and $w_{Ci}$ ($\rho = -0.70$, $p = 0.0034$, $n = 15$). Error bars represent standard errors (SEM). Source data are provided as a Source Data file

curves can be fitted by adjusting these parameters, even without accounting for recruitment difference. Nevertheless, to test if the model captures the effect of recruitment, one can use a realistic set of parameters and test if asymmetric recruitment produces the effects seen during the behavior. We thus looked at the qualitative behavior of the model using a set of parameters obtained in a previous group of experiments[25] by fitting the individual learning curves from 15 mice trained in a task identical to the one used in this study. This parameter set included 15 different values of the initial weights, and core parameters were identical for all mice, which we showed is sufficient to account for inter-individual variability[25]. Based on these parameters, and systematically varying the recruitment values in simulations, we observed that recruitment differences were positively correlated to learning phase duration in the model, similar to the experimental results (Fig. 4b). Also, as illustrated when neuronal recruitment is doubled for one of the two stimuli, the model reproduced two other experimental observations. First, the delay phase was not significantly influenced by recruitment (Fig. 4c). Second, the variability of the learning phase duration was much stronger when S− recruited less activity than S+ (Fig. 4c, d). Thus, without any tuning, the model qualitatively reproduced the complexity of the experimental dynamics, offering an opportunity to explore possible mechanisms for the complex effects of neuronal recruitment on learning behavior in a precise theoretical framework.

**Learning speed effects depend on initial synaptic strengths**. To understand the origin of inter-individual variability in our simulations, we plotted the learning phase duration against the values of the three initial synaptic weight parameters ($w_{Ce}$ and $w_{Ci}$ for the Ĉ population, and a single initial value $w_S$ for the four weights of the Ŝ+ and Ŝ− population, as in Fig. 4a). First, we

observed that the initial weight between sound-specific neural populations and the decision populations ($w_S$) had no correlation with learning speed duration. This was expected as $w_S$ mostly impacts the delay phase because small initial synaptic weights lead to slow initial learning (multiplicative rule). After the delay phase, sufficient learning has occurred for the sound-specific synaptic weights to increase performance at high speed and $w_S$ does not influence learning speed anymore. Our earlier results[25] showed that $w_S$ is the main determinant of the delay phase duration and is highly variable across mice. These inter-individual $w_S$ variations induced large variations of delay phase duration masking the smaller impact of neuronal recruitment on delay phase in simulations (Fig. 4). This suggests that variability in initial connectivity may explain the independence between cortical recruitment and delay phase duration in behavior.

A second observation was that the learning phase is long when the initial weights $w_{Ce}$ and $w_{Ci}$ (non-specific population Ĉ) are small, but only when S− recruits less activity than S+ (Fig. 4d). In contrast, for large weights, the recruitment differences between S+ and S− have no effect on learning phase (Fig. 4d). Hence, variability in initial connectivity from non-specific representation of the task may explain the larger variability in learning phase duration observed in behavior when cortical recruitment for S− is smaller than recruitment for S+ (Fig. 3e).

To better understand why the initial conditions of $w_{Ce}$ and $w_{Ci}$ gate the influence of neuronal recruitment on learning speed, we plotted the time-course of both excitatory and inhibitory connection weights for four combinations of recruitment and initial weights values (Fig. 5a–d, see also Supplementary Fig. 4 for the same analysis with a smaller recruitment ratio). These plots show that the large prolongation of the learning phase when neuronal recruitment is lower for S− than for S+ and when $w_{Ce}$ and $w_{Ci}$ are small (Fig. 5d) is due to the maintenance throughout

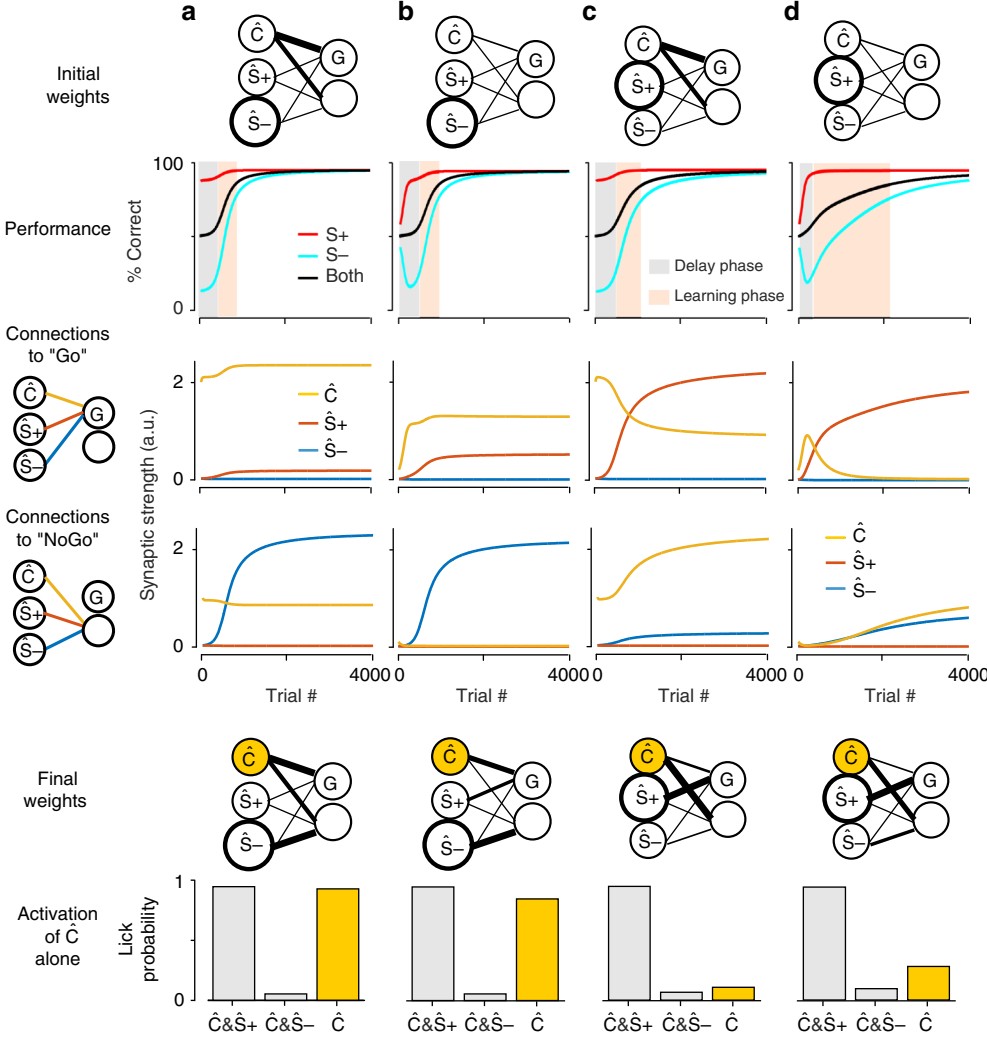

**Fig. 5** The effects of neuronal recruitment in the model are explained by the differential adjustment of synaptic weights during learning. **a** (top) Sketch of the initial synaptic weights and simulated model performance for S+ (red), S− (light blue), and both stimuli (black). (middle) Values of the connections to the excitatory (G) and inhibitory decision populations as indicated by the schematics on the left-hand-side. Yellow: connections from the "common" Ĉ population. Red: connections from the Ŝ+ population. Blue: connections from the Ŝ− population. (bottom) Sketch of the connectivity pattern after and response probability after learning for the S+ (co-activation of Ŝ+& Ĉ) and S− (co-activation of Ŝ− and Ĉ) reinforced stimuli as well as for the common stimulus component alone (Ĉ, yellow). Simulation parameters: $\mathbf{X} = [1; 1; 2]$, $\alpha = 0.01$, $\sigma = 0.6195$, $v = 6$, $w_{Ci} = 1$, $w_{Ce} = 2$, and $w_{S} = 0.01$. **b**. Same as **a**, but with $w_{Ci} = 0.1$, $w_{Ce} = 0.2$. **c** Same as **a**, but with $\mathbf{X} = [1; 2; 1]$. **d** Same as **c**, but with $w_{Ci} = 0.1$, $w_{Ce} = 0.2$

the delay phase of low initial weights from Ŝ− and Ĉ populations to the No-Go population. This impacts the effective speed (multiplicative learning rule) at which correct rejections responses to S− are acquired during the learning phase, flattening the overall learning curve. In contrast, with large initial synaptic weights from the Ĉ population, more rapid S− rejection learning is obtained solely based on the Ĉ common population (Fig. 5c).

When S− recruits more activity, learning is always fast (Fig. 5a, b) because, in all cases, the rate limiting process remains the abolition of licking to the NoGo stimulus (due to learning rule asymmetry, see Supplementary Fig. 3). This process is boosted by strong Ŝ− recruitment. Also, in these conditions, acquisition of the NoGo stimulus is independent of $w_{Ce}$ and $w_{Ci}$, because the Ĉ population drives the Go response. Thus, the complex modulation of learning phase duration by neuronal recruitment is due, in the model, to a non-trivial assignment of the three sensory populations to either Go or NoGo responses, based on neuronal recruitment distribution. Specifically, when the Ŝ− population recruits more activity, it is assigned to NoGo, while the Ĉ population drives the Go response. In contrast, when the Ŝ+

population recruits more activity, it is assigned to the Go response, and in this case, the Ĉ population drives the NoGo response. These different solutions of the binary discrimination problem could be seen as different strategies chosen by the model or eventually the animals during the learning process.

The existence of multiple solutions to the task provides a testable, general prediction of reinforcement learning models which use population activity as a salience parameter (independent of the magnitude of recruitment differences, see analytical arguments in Supplemental Experimental Procedures). The test would be to isolate and drive the neurons corresponding to Ĉ in the brain. Activation of the Ĉ population alone should drive licking when S− recruits more activity, and should not drive licking when S− recruits less activity (Fig. 5, Supplementary Fig. 4).

**Recruitment determines learning strategy**. Testing this prediction was impractical with our sound-based Go/NoGo discrimination protocol, because the neurons encoding information common to S+ and S− trials (Ĉ population) likely code for

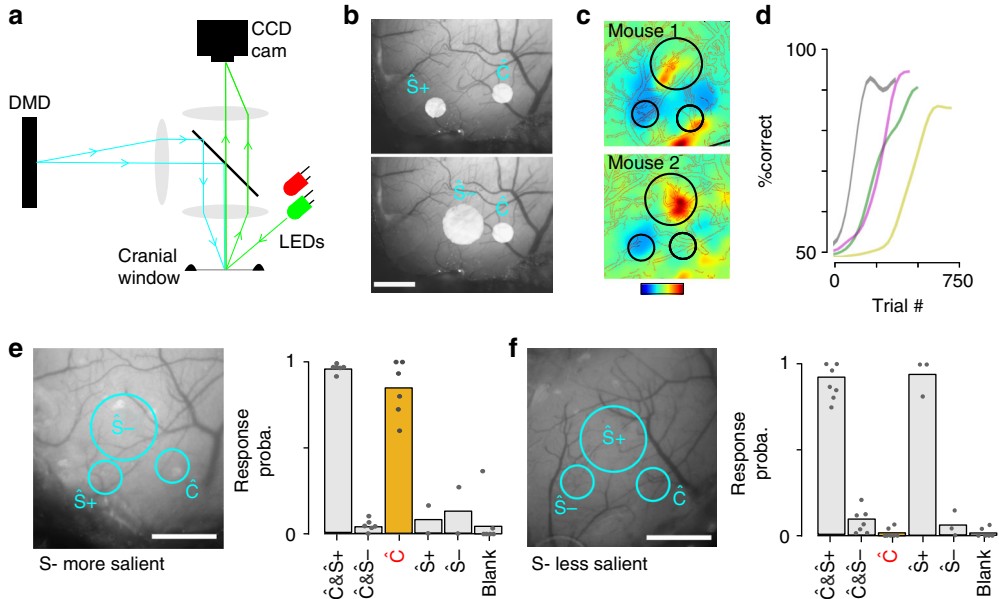

**Fig. 6** Discrimination training of multi-spot optogenetic patterns reveals a choice of learning strategy depending on the level of cortical recruitment. **a** Schematics of the optical setup to project arbitrary 2D light patterns onto the surface of auditory cortex. Blue light patterns from a Digital Micromirror Device (DMD) are collimated and deflected through the objective lens by a beam splitter. The surface of the cranial window can be simultaneously imaged by a CCD camera using external LEDs (green for blood vessel, red for intrinsic imaging). **b** Examples of light patterns used for the discrimination task. Scale bars: 600 μm. **c** Tonotopic maps obtained with intrinsic imaging for two mice, and localization of the three optogenetic stimulation spots in the same high, low, and mid frequency fields. **d** Four example learning curves for the optogenetic discrimination task. Gray: smaller Ŝ−; color: larger Ŝ−. **e** (left) Light pattern in the task in which the S− stimulus has higher level of cortical recruitment (Ŝ− >Ŝ+). Ĉ = common component of S+ and S− stimuli. (right) Response probability for the two learnt target stimuli (Ĉ & Ŝ+vs Ĉ & Ŝ−), for the presentation of the common part of the stimuli alone (Ĉ, yellow), for the specific parts of the stimuli alone (Ŝ+; Ŝ−) and in absence of stimulation (blank). Ten to 15 catch trials per mouse. Mean and individual data points, $n = 6$ mice (except Ŝ+ & Ŝ−, $n = 2$ mice). Gray dots: single animals. **f** Same as **e**, but with Ŝ+ larger than Ŝ−. Mean and individual data points, $n = 7$ mice (except Ŝ+ & Ŝ−, $n = 3$ mice). Scale bars: 600 μm. Source data are provided as a Source Data file

multiple cues, including (i) the overlap of S+ and S− representations and (ii) all cues related to the decision to visit the lick port, and thus cannot be isolated. Therefore, we decided to test the model predictions in an artificial but better controlled experiment in which head-fixed mice had to discriminate optogenetically driven cortical ensembles. We used a custom-made video-projector setup[44] to project precise 2D light patterns through a cranial window placed above the auditory cortex in Emx1-Cre x Ai27 mice (Fig. 6a, b, Supplementary Fig. 5). Using intrinsic imaging, we identified the main tonotopic fields of mouse auditory cortex (Fig. 6c)[33,45] and thereby reliably positioned optogenetic stimulation spots in homologous regions across mice (Fig. 6c). We defined three circular optogenetic stimulation spots out of which we constructed two stimuli. One of the three spots, the Ĉ spot, was common to the two S+ and S− stimuli and the two other spots corresponded to the stimulus-specific neuronal populations Ŝ+ and Ŝ− (e.g. Fig. 6b). Thus, the Go-trial cue consisted of simultaneous activation of Ŝ+ and Ĉ and the NoGo-trial cue consisted of simultaneous activation of Ŝ− and Ĉ. Furthermore, cues common to S+ and S− related to licking port visits in the sound task were eliminated in the head-fixed task design, as mice did not initiate the randomly interleaved trials. Thus the Ĉ spot was the only cue common to Go and NoGo-trials. We doubled the diameter of either the Ŝ+ or the Ŝ− spot to create a difference in cortical recruitment between the two input representations. Electrophysiological measurements of the population firing rate elicited by the small and large spots showed that the recruitment difference between the stronger and weaker stimuli was ~69% (Supplementary Fig. 5), comparable with the population recruitment differences observed for sounds (Fig. 4b; Supplementary Fig. 5). Also, we measured that the large disk activates ~2.5 more neurons than the small disk

(Supplementary Fig. 5). Given that the Ĉ spot has a small diameter, we could evaluate the fraction of cells commonly activated by the S+ and S− stimuli within the cells activated by S+ and S− as $1/(1+1+2.5) \approx 22\%$, similar to the fraction of cells commonly activated by sounds (e.g. 18%, 18%, 23% for sound pairs A–B, A–C, and B–C, s.e.m = 0.3%, binomial distribution). Thus the artificial stimuli, although not identical to sound responses had comparable characteristics.

Mice were then initially trained to obtain a water reward by licking after the coincident activation of the Ŝ+ and Ĉ spot. When 80% performance was reached in this stage, the discrimination training started. Mice kept licking in the presence of the Ŝ+ spot and learned, within hundreds of trials to avoid licking in the presence of the Ŝ− spot (both activated together with Ĉ, Fig. 6d), reaching a steady state performance of 94.7% ± 4.5% correct trials (hit rate 93.9% ± 3.3%, false alarm rate 4.5% ± 1.3%, $n = 8$ mice, see Fig. 6e, f). Importantly, in this task setting, the stringent definition of the common Ĉ population, activated during the initial motivation training, likely resulted in the systematic establishment of strong initial connections for this population at the beginning of the discrimination training, leading to homogenous durations of the learning phase (212 ± 117 trials for the large Ŝ− vs 260 ± 220 for the small Ŝ−, $p = 0.26$, Wilcoxon rank-sum test, see also Fig. 6d) independent of recruitment (Figs. 4 and 5).

However, once mice had learned the behavioral task, we measured their response to Ĉ activation alone in catch trials that were not rewarded (catch trial probability = 0.1; 15 catch trials per mouse). In the group of mice that had a larger Ŝ− spot, we observed that activation of Ĉ elicited strong licking responses (84% ± 6% response probability, $n = 6$ mice, Fig. 6e). In contrast, in the group of mice that had a larger Ŝ+ spot, activation of Ĉ

elicited no licking response (2% ± 1% response probability, $n = 7$ mice, Fig. 6f). Note that these effects are unlikely to be caused by inhibition of Ĉ when paired with the large Ŝ+ or Ŝ− spot, as no inhibition from the larger spot was observed at the location of Ĉ in calibration experiments (Supplementary Fig. 5). In addition, behavioral responses to Ŝ+ or Ŝ− alone were compatible with our model (Figs. 6e, f and 5). By confirming the model's predictions, these results demonstrate, in a causal manner, that cortical recruitment affects the choice of which stimulus is associated to a particular response. Even if simple in essence, this result shows that cortical recruitment is a parameter influencing learning, in a manner compatible with the role of a salience parameter in reinforcement learning models.

## Discussion

Combining behavioral measurements, large scale two-photon imaging, optogenetics, and theoretical modeling, we have shown that sounds of different quality but equal mean pressure levels can recruit highly variable levels of neuronal activity in auditory cortex, measured as the mean amount of activity in a representative subsample of neurons. We showed that cortical recruitment levels correlate with learning speed effects in a Go/NoGo task as expected if neuronal recruitment corresponds to stimulus salience. Moreover, these effects can be precisely reproduced by a reinforcement learning model of the task. Finally, training mice to discriminate optogenetically evoked cortical patterns, and manipulating these patterns, we showed that neuronal recruitment determines which elements of the cortical representation are selected to drive each conditioned action. This corroborates, in a causal manner, the idea that cortical recruitment is a neuronal correlate of stimulus salience.

Several studies indicate that cortical recruitment can vary across stimuli, even when played at the same sound pressure level[26,28,34]. These discrepancies may have multiple origins. First, it is well known that the mouse cochlea is more sensitive in its middle frequency range[29], which could explain the over-representations of sounds in this frequency range (10–30 kHz) in cortex[28]. In this case, cortical recruitment is expected to reflect recruitment throughout the auditory system, making it a good proxy for sound salience independent of whether the discrimination task requires auditory cortex[46–48] or does not require it[49–51]. But a second source of recruitment differences may be the nonlinearities of cortical representations[34,52–54]. For example, a recent study suggested that cortical response patterns can be invariant to changes in intensity[55]. In this case, cortical recruitment should also depend on the higher level features composing sound representations and on how broadly these features are represented in cortex.

The idea that the amount of neuronal activity recruited by a stimulus influences behavior has been proposed in different contexts. For example, several studies indicate that attention can boost neuronal firing associated to behaviorally relevant stimuli[6,56,57] and thereby make them more discriminable from other stimuli[58]. Also, several theoretical studies have proposed that attention impacts learning[59,60] and some reinforcement learning models can account for such effects by dynamically weighting stimuli according to their predictive relevance[61]. It will be an interesting research avenue to analyze the relative contribution of bottom-up sound encoding and attentional top-down mechanisms to the level of cortical recruitment. Earlier reports, using direct microstimulation of the cortex, showed that low levels of neuronal recruitment can impact detection probability[62,63]. Here, we show that neuronal recruitment for stimuli that are well beyond detection threshold still impact the learning process. Even if such effects are predicted by the Rescorla

and Wagner model[3], capturing their details requires a refinement of the original model. In particular, we had to introduce a more realistic multiplicative learning rule which renders learning speed not only dependent on neuronal recruitment, but also on the current synaptic strength. This property has important consequences. First, it introduces variability in the relationship between recruitment and learning speed, through large inter-individual variations of the initial weights for the synapses involved in the task. Second, the multiplicative rule makes learning speed proportional to the product of neuronal recruitment and connectivity, allowing for more robustness, by compensating weak neuronal recruitment with stronger initial connections (see Supplemental Experimental Procedures). In our experiments, this phenomenon tends to stabilize learning speed, explaining why neuronal recruitment does not always impact the learning phase duration, except for particular initial conditions for which compensation occurs too slowly (Figs. 3–5).

Strong pre-established connections can lead to fast learning for specific stimuli with innate meaning. In this study, we have shown an effect of cortical recruitment on learning speed for five different pairs of sounds which had no particular meaning to the animal. This does not exclude the possibility that some sounds, in particular, sounds with learnt or innate meaning would show a different relationship. For example, pup calls are extremely salient to mothers but trigger little activity in cortex[64]. This could be due to strong pre-existing wiring between cortical neurons responding to pup calls compensating for limited recruitment. So, even if, as we show, cortical recruitment plays a role in the salience of a sound, general theories of salience should also account for potential a priori meaning of stimulus, via pre-existing connections, as simple extensions of our model would suggest, or via more complex cognitive processes assigning value to the sounds.

The complex dynamical phenomena described in our study make learning speed measurements a more complicated proxy for stimulus salience than the overshadowing protocol which relies on steady state behavior, after the dynamical phase of the association. However, it allows the comparison of salience for stimuli from the same sensory modality. As our extended multiplicative model only diverges from the Rescorla-Wagner model for the transient dynamical part of the association process, it reproduces overshadowing effects[25], and can also predict how elements of sensory representations are assigned to different conditioned responses in a more complex task setting. Here, by conditioning mice to compound stimuli composed of multiple optogenetically activated neuronal ensembles (Fig. 6), we show, in line with reinforcement learning models, that the brain establishes its stimulus discrimination strategy based on the amount of activity recruited by the different subpopulations representing the stimuli.

## Methods

**Animals**. All mice used for imaging and behavior were 8–16 weeks old C57Bl6J and GAD2-Cre (Jax #010802) × RCL-TdT (Jax #007909) mice. Mice used for optogenetics were 8–16 weeks old males and female obtained by crossing homozygous Emx1$^{IRES-cre}$ (Jax #005628) mice with Ai27 (Jax # 012567) mice to obtain expression of Td-Tomato-tagged channelrhodopsin (ChR2) in excitatory neurons of the cortex. All animal were group housed. All procedures were approved by the Austrian laboratory animal law guidelines (Approval #: M58/02182/2007/11; M58/02063/2008/8) and the French Ethical Committee (authorization 00275.01).

**Two-photon calcium imaging in awake mice**. At least 3 weeks before imaging, mice were anaesthetized under ketamine medetomidine. The right masseter was removed and a large craniotomy (~5 mm diameter) was performed above the auditory cortex. We then performed three to five injections of 200 nL (35–40 nL min⁻¹), rAAV1.Syn.GCaMP6s.WPRE virus obtained from U. Penn Vector Core (Philadelphia, PA) and diluted 10 times. The craniotomy was sealed with a glass window and a metal post was implanted using cyanolite glue and dental cement. At least 3 days before imaging, mice were trained to stand still, head-fixed under the microscope for 20–60 min per day. Then mice were imaged one to two hours per day. Imaging was performed using a two-photon microscope (Femtonics,

Budapest, Hungary) equipped with an 8 kHz resonant scanner combined with a pulsed laser (MaiTai-DS, SpectraPhysics, Santa Clara, CA) tuned at 920 nm or 900 nm depending on the experiments. Images were acquired at 31.5Hz. All sounds were delivered at 192 kHz with a NI-PCI-6221 card (National Instrument) driven by Elphy (G. Sadoc, UNIC, France) through an amplifier and high frequency loudspeakers (SA1 and MF1-S, Tucker-Davis Technologies, Alachua, FL). Sounds were calibrated in intensity at the location of the mouse ear using a probe microphone (Bruel&Kjaer). In a first experiment, we played three 70 ms complex sounds at 73 dB SPL preceded by two 50 ms 4 kHz pure tones (inter-tone interval: 375 ms) sounds as in the behavioral task. The three sounds were played in a random order and repeated 30 times. In a second experiment, we played white noise sounds ramping -up or -down in intensity between 60 dB and 85 dB SPL during 2 s. The ramps were repeated 20 times. In a third experiment, we played 20 repetitions of white noise sounds modulated in intensity at 1 and 20Hz.

**Data analysis**. Data analysis was performed using Matlab and Python scripts. Motion artifacts were first corrected frame by frame, using a rigid body registration algorithm. Regions Of Interest were selected using a semi-automated hierarchical clustering algorithm based on pixel covariance over time as described in[30] (see detailed method below). Neuropil contamination was subtracted[65] by applying the following equation: $F_{true}(t) = F_{measured}(t) - 0.7\ F_{neuropil}(t)$, then the change in fluorescence ($\Delta F/F_0$) was calculated as $(F-F_0)/F_0$, where $F_0$ is estimated as the minimum of the low-pass filtered fluorescence over ~40 s time windows period. To estimate the time course of firing rate, the calcium signal was temporally deconvolved using the following formula: $r(t) = f'(t) + f(t)/\tau$ in which $f'$ is the first time derivative of $f$ and $\tau$ the decay constant set to 2 s for GCaMP6s. For the complex sounds, the population response was computed as the mean deconvolved signal across all neurons from sound onset to 500 ms after sound offset. For the ramps, because the behavioral discrimination response typically occurs within few hundreds of milliseconds after the ramp onset[34], the mean population response was evaluated from 0 to 500 ms after sound onset. To estimate the discriminability of two sounds based on cortical population responses, linear Support Vector Machine classifiers were trained to discriminate population activity vectors obtained from 20 presentations of each sound (training set), and were tested on activity vectors obtained for 10 independent presentations of the same sounds (test set).

**Patterned optogenetics and intrinsic imaging**. To flexibly activate different activity patterns in the mouse auditory cortex, we used a computer driven (VGA input) video projector (DLP LightCrafter, Texas Instruments) which includes a strong blue LED light source (460 nm) and from which we have removed the objective. To project a two-dimensional image onto the auditory cortex surface (Fig. 6a, b), the image of the micromirror chip is collimated through a 150-mm cylindrical lens (Thorlabs, diameter: 2 inches) and focused through a 50-mm objective (NIKKOR, Nikon). Imaging of the cortex at the focal plane is obtained by side illumination with a green (525 nm, blood vessels) or far red (780 nm, intrinsic imaging) LED. The light collected by the objective passes through a dichroic beamsplitter (long pass, >640 nm, FF640-FDi01, Semrock) and is collected by a CCD camera (GC651MP, Smartek Vision) equipped with a 50-mm objective (Fujinon, HF50HA-1B, Fujifilm). Note that the image projected to the cortical surface corresponds to a narrow cone of light extending below the surface and potentially activating ChR2 expressing neurons throughout the cortical depth. Intrinsic imaging was performed in isoflurane anesthetized mice (1.1% delivered through SomnoSuite, Kent Scientific). To compute intrinsic signal maps we divided the red light image of the cortical surface after the onset of a stimulation (average over 2 s) with 2-s-long pure tones (4, 8, 16 and 32 kHz) by the mean image immediately before stimulus onset.

**Calibration of optogenetic**. For calibration of optogenetic stimulation, a small aperture was drilled in the cranial window with a diamond-coated dental drill during isoflurane anesthesia. 30 min after surgery, $4 \times 8$ silicon probes (Neuronexus) were implanted at a ~35° angle in auditory cortex in the awake head-fixed mouse. Recordings were performed at three different depths (400, 600, and 800 μm) using a pre-amplifier and multiplexer coupled to a USB acquisition card (Intan Technologies). Sounds and light stimulations were randomly presented at 2.5 s and each repeated 10 times. Single unit spikes were detected and sorted from multi-unit spikes using the Phy Suite (https://github.com/kwikteam/phy). Light stimuli consisted of small and large disk (360 μm and 720 μm diameter) presented at different positions of a two-dimensional grid centered on the probe location (Supplementary Fig. 5). For small disks, the grid included $5 \times 8$ locations with a regular $\Delta x = \Delta y = 360$ μm spacing. For large disks, the grid included $3 \times 4$ locations with a regular spacing of 480 μm. Population firing rate elicited by a single disk was estimated as the integral $\sum_{Locations} r_{x,y} \Delta x \Delta y$ of the responses $r_{x,y}$ to the disk centered at locations that were within the extent of the smaller of the two grids (large disks: all locations, extent $1440 \times 1920$ μm; small disk: $4 \times 5$ locations, spanning $1440 \times 1800$ μm). The integrals for big and small disks were normalized by $\Delta x_0 \Delta y_0 = 360 \times 360$ μm². The overall fraction of responsive neurons over the extent of the smaller grid (see firing rate) was computed as $\left( \sum_{x,y} N_{responsive\_cells}(x,y) \right) / (N_{locations} \times N_{cells})$,

where $N_{responsive\_cells}(x, y)$ is the number responsive cells at each disk location, $N_{location}$ is the number of locations and $N_{cells}$ is the number of recorded cells.

**Go/NoGo discrimination behavior**. Mice were water-deprived and trained daily for 200–300 trials. Mice first performed 4 habituation sessions to learn to obtain a water rewards (~5 μL) by licking on a spout over a threshold after the positive stimulus S+. After habituation, the fraction of collected rewards was ~80%. The learning protocol then started in which mice also received a non-rewarded, negative sound S− for which they had to decreasing licking below threshold to avoid an 8 s time-out. For the freely moving complex sound discrimination, S+ and S− sounds consisted of two 4 kHz pips (50 ms) followed by one of the three 70 ms complex click shown in Fig. 1a. The interval between the offset and onset of the pips and click was 375 ms. Licking was assessed 0.58 s after the specific sound cue in a 1-s long window by an infrared beam at the spout. For the intensity ramp discrimination, licking was assessed in a 1.5-s window after sound offset. In both cases, licking was considered above threshold if the infrared beam in front of the licking tube was broken during 75% of the measurement time-window. Positive and negative sounds were played in a pseudorandom order with the constraint that exactly 4 positive and 4 negative sounds must be played every 8 trials. For learning curves, performance was measured as the fraction of correct positive and correct negative trials over bins of 100 trials. For the optogenetically driven, head-fixed discrimination task, the S+ and S− stimuli were each composed of two disks of blue light (465 nm) flashing at 20Hz for 1 s. One of the two disks (noted Ĉ) was common to S+ and S− stimuli, the other disk was condition-specific. The three disk locations were chosen in similar tonotopic locations across mice based on intrinsic imaging maps. They were precisely re-positioned for every training session using an automated registration procedure based on blood-vessel patterns. A strong masking light was used to prevent the animal from using visual cues in the task. In one set of mice, the disk specific to S− was 720 μm in diameter, while the S+ specific disk was 360 μm in diameter. In the other set of mice, sizes of the specific disks were swapped. Head-fixed mice performed 200–300 trials per day with an inter-trial interval randomized between 3 and 7 s. Individual licks were detected through an electric circuit connecting the mouse and the lick tube. Then, each trial was started only if the mouse was not spontaneously licking for at least 3 s (in addition to the inter-trial interval). Mice were first trained to respond to the S+ stimulus by producing at least 3–5 licks (depending on the mouse) to get the 5 μL water reward. When the mouse could collect more than 80% of the rewards, the S− stimulus was introduced. Licking above threshold after S− was punished with a 7-s timeout.

**Reinforcement learning model**. The model has been described extensively in a previous publication[25]. In short, it is composed of three sensory units (Ŝ+, Ŝ−, and Ĉ, representing populations of neurons) whose activity described by a three-dimensional vector **X** and which are connected to a simple decision circuit (Fig. 4a). Cortical recruitment is modeled by changing the firing value of the sound units. When the S− stimulus recruits less activity than the S+ stimulus, the input vectors are: $\mathbf{X}_{S+} = [1\ 0\ 2]$ or $\mathbf{X}_{S-} = [1\ 1\ 0]$. When S− recruits more activity than S+, the input vectors are: $\mathbf{X}_{S+} = [1\ 0\ 1]$ or $\mathbf{X}_{S-} = [1\ 2\ 0]$.

The decision circuit is composed of all-or-none response unit ($y = 0$ or 1) which linearly sums the three sensory inputs (representing synaptic populations) under the form of three direct excitatory connections and of a graded feed-forward inhibition from a virtual inhibitory unit in fact equivalent to three direct inhibitory connections. The output of model is described by a single equation for the decision unit:

$$y = \theta(\mathbf{W}_E.\mathbf{X} - \mathbf{W}_I.\mathbf{X} - \xi) \tag{1}$$

in which $\theta$ is the Heaviside step function. $\mathbf{W}_E$ and $\mathbf{W}_I$ are three-dimensional positive vectors describing the excitatory synaptic weights from the sensory units to the decision and inhibitory units respectively. The variable $\xi$ is a Gaussian random noise process of unit variance which models the stochasticity of behavioral choices.

Based on the action outcome ($R = 1$ for a reward, $R = -1$ for no reward), the learning rule for the synaptic weights is implemented as:

$$\delta\mathbf{W}_E = \alpha\mathbf{W}_E \odot f[R - \sigma(\mathbf{W}_E - \mathbf{W}_I).\mathbf{X}]y\mathbf{X} \tag{2}$$

$$\delta\mathbf{W}_I = -\alpha\mathbf{W}_I \odot f[R - \sigma(\mathbf{W}_E - \mathbf{W}_I).\mathbf{X}]y\mathbf{X} \tag{3}$$

in which $\odot$ is the Hadamard (element-wise) product implementing the multiplicative rule, $y\vec{x}$ is a Hebbian term, $\alpha$ is the learning rate and $\sigma$ is a parameter related to the noisiness of the model and setting its asymptotic performance. To account for the faster improvement of performance for rewarded as compared to non-rewarded trials, positive expectation errors are more strongly weighted than negative ones, thanks to the asymmetric function $f[u] = u$ if $u \leq 0$ and $f[u] = vu$ if $u > 0$. The parameter $v$ is typically >1, consistent with the activity of basal ganglia dopaminergic neurons in mice[66] and monkeys[67] coding for reward expectation error.

As described above, the equations of the model are stochastic due to the Gaussian random noise process $\xi$. To compute the response probability estimates plotted throughout the study, we used a previously established probability equation[25], valid for learning dynamics much slower than fluctuations (ergodic

approximation). The probability to make a lick response given the input vector $\mathbf{X}_{s+}$ or $\mathbf{X}_{s-}$ is:

$$p_{S+|S-} = p(y = 1|\mathbf{X} = \mathbf{X}_{S+|S-}) = \frac{1}{2}\left(1 + \mathrm{erf}\left(\frac{\Delta\mathbf{W}.\mathbf{X}_{S+|S-}}{\sqrt{2}}\right)\right) \quad (4)$$

where $\Delta\vec{w} = \vec{w}_E - \vec{w}_I$ represents now the average observed values of difference between the excitatory and inhibitory connections. In addition, the plasticity equations become:

$$\delta\mathbf{W}_E = \frac{\alpha}{2}\mathbf{W}_E \odot \left(f[1 - \sigma\Delta\mathbf{W}.\mathbf{X}_{S+}]p_{S+}\mathbf{X}_{S+} + f[-1 - \sigma\Delta\mathbf{W}.\mathbf{X}_{S-}]p_{S-}\mathbf{X}_{S-}\right) \quad (5)$$

$$\delta\mathbf{W}_I = -\frac{\alpha}{2}\mathbf{W}_I \odot \left(f[1 - \sigma\Delta\mathbf{W}.\mathbf{X}_{S+}]p_{S+}\mathbf{X}_{S+} + f[-1 - \sigma\Delta\mathbf{W}.\mathbf{X}_{S-}]p_{S-}\mathbf{X}_{S-}\right) \quad (6)$$

**Statistical tests.** Unless otherwise specified, all quantifications are given as mean ±standard error (SEM). To statistically assess the differences between paired measurements (e.g. activity for two different sounds elicited in the same neuronal populations) we used the non-parametric Signed test. To compare two sets of measurements (e.g. delay and learning phase duration for two groups of mice) we used the non-parametric Wilcoxon rank sum test. Assessment of the differences in the fraction of responsive neurons for different sounds was done with the $\chi^2$ test which evaluates differences in the distributions of two binary variables. All tests are two-sided. No data was excluded from the analysis sample.

**Reporting summary.** Further information on experimental design is available in the Nature Research Reporting Summary linked to this article.

## Data availability
All datasets, analysis software, and codes for running the simulations of our model are freely available on the Dryad Digital Repository (http://datadryad.org/) https://doi.org/10.5061/dryad.47h8t87 or on https://www.bathellier-lab.com/downloads. The source data underlying Figs. 1c, 3d, and 6e, f are provided as a Source Data file.

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

## Acknowledgements

We thank K. Kuchibhotla, E. Harrell, and M. Stüttgen for comments on the manuscript, P. Pindi, S. Sikirić, and L. François for help with behavioral and imaging experiments. We thank the GENIE Project, Janelia Farm Research Campus, and Howard Hughes Medical Institute for GCAMP6s constructs. This work was supported by the Agence Nationale pour la Recherche (ANR "SENSEMAKER"), the Fyssen foundation, the DIM "Region Ile de France", the Marie Curie Program (CIG 334581), the International Human Frontier Science Program Organization (CDA-0064-2015), by the Fondation pour l'Audition (Laboratory grant), the École Doctorale Frontières du Vivant (FdV)—Programme Bettencourt (support to A.K.), the European Research Council (ERC CoG DEEPEN), the DIM Cerveau et Pensée and Ecole des Neurosciences de Paris Ile-de-France (ENP, support to S.C.) and the Deutsche Forschungsgemeinschaft (DFG CRC1080/2).

## Author contributions

B.B. and S.R. designed the study. A.K., S.C., and B.B. performed and analyzed the imaging experiments. B.B. and J.B. performed the modeling. A.D and B.B. performed and analyzed behavioral experiments. Z.P. designed the patterned optogenetic setup, S.C. and P.P. performed the optogenetic experiments. T.D. designed software for data analysis and behavior. B.B. and S.R. wrote the manuscript with comments from all authors.

## Additional information

**Competing interests:** The authors declare no competing interests.

