## [Peer Review File · Nature Communications]

Editorial Note: During the final round of review, the editors sought advice from Reviewer 3 regarding the authors' responses to the remaining concerns of Reviewer 1 and guided the authors to revise the manuscript accordingly.

Reviewers' Comments:

Reviewer #1:

Remarks to the Author:

This is an extremely interesting paper tackling the relationship between cortical activity, learning, and salience. The paper leverages the Bathellier lab's strong combination of behavioral methods, two-photon imaging, optogenetics, and modeling to probe the neural underpinnings of stimulus salience. This paper will be of wide interest to neuroscientists and psychophysicists.

The authors want to show a salience -> cortical recruitment link using two core links in the paper: cortical recruitment -> learning rate, and learning rate -> salience.

The authors show experimentally a link between cortical recruitment and learning rate - lesser cortical activity is recruited by Sound A, and it takes mice longer to learn to associate Sound A to NoGo response. Some questions remain about the interpretation of these experiments, and whether this association only occurs in the context of discrimination training.

The second connection between learning rate and salience is via a model, and is more tenuous. The authors claim that because an overshadowing paradigm cannot be used in the auditory domain (sounds could perceptually fuse), learning rate is used as a metric of salience. It is unclear if this fact (learning rate correlates with salience) is a pre-existing finding arising from independent experiments (in which case citations/discussion is necessary), or if this is one the authors are proposing. If this is being proposed here, then the logic may be circular, because the authors operationally define salience as higher cortical recruitment, which is what they set out to prove.

The optical sculpting experiments are reminiscent of experiments in the olfactory system, but are very nicely done, the testing paradigm is clever, and this will be a valuable tool for the community. In all, the technical aspects of the paper are strong, and I'm overall positive about the impact of this paper. But I would recommend a major revision to get the logic straightened out.

Major concerns

1. The authors place a lot of emphasis on cortical recruitment's relationship to sound intensity. There are two main questions here:

1a) Although the authors have normalized the r.m.s. levels of the three sounds, the mouse audiogram will impose a further frequency-dependent attenuation of the sounds. A sound composed of predominantly low frequencies, for example, will elicit less cortical activation (or for that matter, auditory nerve activation), than a sound composed of frequencies at the mouse's audiometric "sweet spot" of ~16-40KHz. Thus, it is not surprising that Sound A, which appears to be of predominantly low frequency, elicited two-fold less activity than the other two sounds and activates fewer neurons (page 6, line 116-120).

1b) The second issue is that some frequencies are over-represented mouse cortex (again, in the 16-40 KHz) range, see recent work from labs of Dan Polley, Patrick Kanold, Troy Hackett. It is expected that sounds that have energies at these frequencies will activate larger cortical areas, and a larger number of neurons, than sounds that are predominantly composed of frequencies that are not as over represented. Again, this seems to be a key difference between Sound A and Sounds B,C. A solution to this would be to acquire the tonotopic map, either at single-cell or transcranial levels, and normalize the "fraction of neurons activated" metric to the area of cortex you would expect to activate in the first place.

It is difficult to judge the extent to which frequency content plays a role because the spectrograms in Fig. 1A are not adequately labeled on the y-axis. Is it log or linear scale? Could you show the overall spectrums of Sounds A, B, C so that other sound quality differences can be appreciated better?

This is all to say that the 'discrepancy' between cortical recruitment and physical intensity is to be expected. The sentences reg. this in the abstract, the first Results sub-heading, and authors summary of the first part of the results (Page 7, Lines 156-160) are along expected lines. Perhaps its what the authors meant to convey, but the writing could be better – there's nothing to be intrigued about the fact that different equal intensity sounds elicit different amounts of cortical activity. What matters is that you have chosen sounds that elicit different levels of cortical activity for the experiment.

A couple of references on cortical recruitment at different intensities:

Wong and Schreiner (2003) "Representation of CV-sounds in cat primary auditory cortex..." shows different patterns/extents of cortical recruitment at the same intensity for different sounds.

Sadagopan and Wang (2008) "Level invariant representation of sounds by populations of neurons..." proposes that cortical recruitment patterns may be intensity-independent for a given sound.

2) Page 9, line 184: The authors measure salience through learning speed. Have other studies used learning speed to measure salience, or is this your proposal? Please provide references if this is an established metric or correlate of salience.

3) The authors mention learning rate as a metric of salience in the discrimination task, but do not show learning rates for just sounds A, B, and C as S+ in the initial conditioning phase. The authors characterize this as a pre-training phase aimed at raising motivation (Page 9, Line 186). But I think this could be an important control – if a stimulus drives lesser cortical activity creating a lower SNR neural signal to guide behavior, one might expect to see a slower learning rate for S+ acquisition as well. Could the authors please show these data?

4) Page 3, line 194: the authors state that "learning speed depends more on the salience of the S- than of the S+ sound. The relative saliences...". This is stated as fact, but is this actually a hypothesis? In the next line, the authors state "If X recruits less activity than Y..." from which it seems you have implicitly assumed that cortical recruitment = salience. But isn't this what you want to show with the paper? Something about the logic is off here. The interchangeable use of "salience" and "cortical recruitment" at this point in the manuscript is not appropriate. On a related note, I would suggest that the authors do not use the S- < S+ terminology in Figure 3. Instead, could the authors use something easier to follow, such as "Sound A (S-), Sound C (S+)" for "S-<S+".

5) Page 12, line 255: the authors state "Most importantly here, the model has an explicit salience parameter, that corresponds to the activity of the S+ and S- populations". Please clearly define which parameter you are referring to here – is it the 0.5 or 2x ratio of activities that you use? From the supplement, it appears that you have once again assumed cortical recruitment=salience, which might make the logic circular.

6) Fig. 5's legend makes a reference to learning "strategy", but its not clear why at this point. I would take "strategy" to mean a decision by the animal that following one set of heuristics works better than others for task performance. You could define this better in the text.

7) Page 20, line 443: This section to me is somewhat naive. There is a rich literature on different levels or patterns of recruitment for different sounds, and the authors' earlier study is one of them.

8) On a more philosophical note, consider the pup-retrieval behavior in mice. The pup distress calls,

owing to their very high frequencies, likely recruit smaller extent of cortical activation than many other stimuli. Yet, pup distress calls are probably highly salient to dams (experienced moms). Wouldn't this be a counter-example to the claim that salient stimuli evoke more cortical activation? In general, I would like to see a brief discussion of alternative models.

Other concerns:

1) Zhaoping Li (2002) "A Saliency map in primary visual cortex" proposes a model that saliency monotonically correlates with V1 neuron firing rates and is mediated by layer 2/3 horizontal connections, and is a highly relevant reference that must be acknowledged.

2) Figure 1A: I suggest moving the labels 'A', 'B' and 'C' to the top of the spectrograms, and the axis labels to the bottom. If the sounds are 70 ms long, then 0 and 70 should line up with the edges.

3) Page 6, line 133,135: percent "%" signs are incorrect.

3) Fig. 3b legend: individual learning curves are only shown for 4 mice, not 6.

4) Fig. 3b,c: The x axis labels "1000 trials" are a bit confusing.

5) In Fig. 5, could you mark, as in Fig. 3, the delay and learning phases?

6) Line 676: Do you mean the U. Penn. viral vector core?

Reviewer #2:

Remarks to the Author:

This paper, building on earlier work by Bathellier et al (2013), finds support for a multiplicative model of reinforcement learning in which stimulus "saliency" (activation of the stimulus representation) influences learning speed after a delay period. The existence of a delay period, as pointed out in the earlier work, supports a multiplicative model in which learning dynamics depend on the synaptic weights. The authors find neurophysiological support for their model using an analysis of cortical recruitment and an elegant optogenetic manipulation.

Overall, I thought this was an interesting and well-written paper. The model is simple but makes non-trivial predictions that depart in important ways from earlier models.

Major comments:

It wasn't clear whether the decoding results shown in Fig. 2f were cross-validated (i.e., was the classifier evaluated on a held-out test set?).

The authors are worried about overfitting the model (p. 12), but I don't really understand why (of course, overfitting is an issue for all models, but I don't see anything special about this case). I think it would be useful for the authors to show that they can capture relationship between recruitment and learning shown in Fig 3, which relies on an analysis of individual differences in model parameter estimates.

If the authors want to claim homogenous durations of the learning phase (p. 18), then this should be quantified in some way.

The authors have chosen to view "saliency" in terms of stimulus activation. This is fine, but it might be worth discussing other views of stimulus saliency in the perceptual and reinforcement learning literatures. For example, saliency is sometimes conceptualized in terms of neuronal gain control. In the RL literature, there is a burgeoning discussion of attention (see for example work by Niv et al 2015) in which stimuli get weighted based on their predictive relevance. There is of course also a rich literature in animal learning (going back to classic works by Pearce and Hall 1980 and Mackintosh 1975) on how attention affects learning.

Minor comments:

typo, p. 4: ", ,"

p. 31, lines 779-781: missing math symbols

Fig 3b: caption says there are learning curves for 6 individual mice, but I only see 4. Relatedly, for 3c it says that there is $n=6$ for each curve, but then for 3d it says $n=4$.

Fig 3d: computing a correlation with 4 data points is somewhat dubious.

Fig 3e: statistical quantification of the claims made about these results is needed.

p. 19: "cortical recruitment affect" -> "cortical recruitment affects"

typo, Supplement, p. 2: "the 2D the plane"

Reviewer #3:

Remarks to the Author:

The authors present behavioral and physiological experiments that suggest that the number of cortical neurons that are activated in auditory cortex by a particular sound reflect the speed in which this sound can be learned as the no-go stimulus in an auditory task. The learning in this task tends to occur in two phases, a first phase where animals respond to the go and the no go stimulus and a second phase where the animal gradually improves by slowly learning to refrain from licking in presence of the no-go stimulus. The authors used the data from the number of neurons recruited to fit a previously published reinforcement learning model that reproduces this behavior, in that the recruitment is proportional to the speed of learning on the ramp phase and does not correlate with the duration of the initial phase. The authors then use an optogenetic training paradigm to show that the recruitment is the crucial parameter and show that a neutral stimulus triggers licking depending on its comparative size respect to the size of the go stimulus. If the neutral stimulus is smaller than the go stimulus, it will not trigger licking, but it would trigger licking if the neutral stimulus is larger than the go-stimulus. This effect can be explained by their reinforcement learning model.

However, there are serious concerns about the validity of the conclusions of this study given the extremely small sample size.

Major concerns

1) The authors tested only 3 sounds and correlated the cortical recruitment with the behavioral saliency. Sounds can differ in many multiple dimensions and cortical recruitment might have correlated by chance with the relevant variable. With a sample of 3, it is hard to reach a conclusion. The authors should increase this number significantly and provide a justification for the number of

sounds used.

2) The optogenetic manipulation recruitment is very different from the recruitment of cortical cells by sounds. The sounds recruit interleaved population of neurons and optogenetics create topologically separated populations of neurons. This strong local stimuli might recruit inhibitory population of neurons in different way than sounds affecting the validity of the model. In addition, the more salient sound recruits at least 400% more neurons than the less salient stimuli, whereas as the more salient sound recruits 10% more neurons than the least salient sound. The authors should at least use optogenetic stimuli that better reflect the magnitude of recruitment differences of actual sounds.

Reviewer #1 (Remarks to the Author):

This is an extremely interesting paper tackling the relationship between cortical activity, learning, and salience. The paper leverages the Bathellier lab's strong combination of behavioral methods, two-photon imaging, optogenetics, and modeling to probe the neural underpinnings of stimulus salience. This paper will be of wide interest to neuroscientists and psychophysicists.

The authors want to show a salience → cortical recruitment link using two core links in the paper: cortical recruitment → learning rate, and learning rate → salience.

The authors show experimentally a link between cortical recruitment and learning rate – lesser cortical activity is recruited by Sound A, and it takes mice longer to learn to associate Sound A to NoGo response. Some questions remain about the interpretation of these experiments, and whether this association only occurs in the context of discrimination training.

The second connection between learning rate and salience is via a model, and is more tenuous. The authors claim that because an overshadowing paradigm cannot be used in the auditory domain (sounds could perceptually fuse), learning rate is used as a metric of salience. It is unclear if this fact (learning rate correlates with salience) is a pre-existing finding arising from independent experiments (in which case citations/discussion is necessary), or if this is one the authors are proposing. If this is being proposed here, then the logic may be circular, because the authors operationally define salience as higher cortical recruitment, which is what they set out to prove.

The optical sculpting experiments are reminiscent of experiments in the olfactory system, but are very nicely done, the testing paradigm is clever, and this will be a valuable tool for the community. In all, the technical aspects of the paper are strong, and I'm overall positive about the impact of this paper. But I would recommend a major revision to get the logic straightened out.

We thank the referee for her/his encouraging evaluation. We have thoroughly revised the manuscript according to her/his recommendations, see details below.

Major concerns

1. The authors place a lot of emphasis on cortical recruitment's relationship to sound intensity. There are two main questions here:

1a) Although the authors have normalized the r.m.s. levels of the three sounds, the mouse audiogram will impose a further frequency-dependent attenuation of the sounds. A sound composed of predominantly low frequencies, for example, will elicit less cortical activation (or for that matter, auditory nerve activation), than a sound composed of frequencies at the mouse's audiometric "sweet spot" of ~16-40KHz. Thus, it is not surprising that Sound A, which appears to be of predominantly low frequency, elicited two-fold less activity than the other two sounds and activates fewer neurons (page 6, line 116-120).

1b) The second issue is that some frequencies are over-represented mouse cortex (again, in the 16-40 KHz) range, see recent work from labs of Dan Polley, Patrick Kanold, Troy Hackett. It is expected that sounds that have energies at these frequencies will activate larger cortical areas, and a larger number of neurons, than sounds that are predominantly composed of frequencies that are not as over represented. Again, this seems to be a key difference between Sound A and Sounds B,C. A

solution to this would be to acquire the tonotopic map, either at single-cell or transcranial levels, and normalize the “fraction of neurons activated” metric to the area of cortex you would expect to activate in the first place.

It is difficult to judge the extent to which frequency content plays a role because the spectrograms in Fig. 1A are not adequately labeled on the y-axis. Is it log or linear scale? Could you show the overall spectrums of Sounds A, B, C so that other sound quality differences can be appreciated better?

This is all to say that the ‘discrepancy’ between cortical recruitment and physical intensity is to be expected. The sentences reg. this in the abstract, the first Results sub-heading, and authors summary of the first part of the results (Page 7, Lines 156-160) are along expected lines. Perhaps its what the authors meant to convey, but the writing could be better – there’s nothing to be intrigued about the fact that different equal intensity sounds elicit different amounts of cortical activity. What matters is that you have chosen sounds that elicit different levels of cortical activity for the experiment.

A couple of references on cortical recruitment at different intensities:

Wong and Schreiner (2003) “Representation of CV-sounds in cat primary auditory cortex...” shows different patterns/extents of cortical recruitment at the same intensity for different sounds.

Sadagopan and Wang (2008) “Level invariant representation of sounds by populations of neurons...” proposes that cortical recruitment patterns may be intensity-independent for a given sound.

We fully agree that difference of cortical recruitment can be explained by many factors, including the frequency range involved, but not only, as shown by our two white noise sound pairs which differs only in their intensity modulation time course (note that we have added another sound pair - AM modulated white noise at 1Hz vs 20Hz - to answer a comment from referee 3). To clarify this, we have:

- improved labeling in Fig 1 to show the log scale, and plotted power spectrum.

- improved abstract and results in order to make clear that it is expected that different spectral and temporal features are not equally represented, citing Wong 2003. The interesting study by Sadagopan 2008 is now cited in the discussion.

Abstract “Here, we identified sounds of equal intensity and perceptual detectability, which due to their spectro-temporal content recruit different levels of population activity in mouse auditory cortex.”

Results “In order to investigate the relationship between stimulus salience and neuronal recruitment, we first aimed to identify sounds recruiting different amounts of cortical activity. A previous report has shown that complex sounds with different frequency content but equal duration and sound pressure level can recruit population responses of different sizes in cat auditory cortex²⁶. To test if a similar result could be obtained in mice, which would then allow us to experimentally decouple recruitment from physical intensity, we chose three short, complex sounds (70ms duration) containing a large range of frequencies and temporal modulations, but normalized at equal mean pressure level (73dB SPL, Fig. 1a). The three sounds displayed different power spectra in the 10-30kHz range (Fig. 1a) where the mouse ear is most sensitive²⁷⁻²⁹. We thus wondered if this discrepancy was affecting their detectability. To do so, we trained mice to lick on a water port after presentation of each of the three sounds”

2) Page 9, line 184: The authors measure salience through learning speed. Have other studies used learning speed to measure salience, or is this your proposal? Please provide references if this is an

established metric or correlate of salience.

Other studies, in particular Rescorla and Wagner, in their seminal model, have postulated that salience directly influences learning speed. So we now clearly state that we follow this hypothesis.

“precluding measurement of their individual saliences with the classical overshadowing design³⁸. Alternatively, the seminal model of Rescorla and Wagner³ postulates that learning speed follows stimulus salience, and this has not yet been addressed experimentally. We thus decided to confirm if stimulus salience can be estimated through learning speed, using an auditory-cued Go/No-Go task.”

3) The authors mention learning rate as a metric of salience in the discrimination task, but do not show learning rates for just sounds A, B, and C as S+ in the initial conditioning phase. The authors characterize this as a pre-training phase aimed at raising motivation (Page 9, Line 186). But I think this could be an important control – if a stimulus drives lesser cortical activity creating a lower SNR neural signal to guide behavior, one might expect to see a slower learning rate for S+ acquisition as well. Could the authors please show these data?

Indeed, intuitively, the learning rate for sound – reward association should also depend on stimulus salience in this phase. However, in fact, in our freely moving task this is not the case. The reason is that mice have to first poke into the lick port to get the S+ sound. Hence there are two cues that can be associated to the licking action that yields the reward: the nose poke action or the S+ sound. We observed that most mice in this phase lick even in the absence of sound indicating that they actually initially use the nose poke action, and that the sound is at least partially overshadowed. This is exactly why we have introduced in our first study (Bathellier et al., PNAS 2013) the C (common) neural population in the model, which captures the initial association made with the nose poke action.

Here are below as examples the learning curves for the initial training phase (rewarded sound only) for the 1Hz vs 20Hz white noise sounds, and for the up and down ramping sounds. Note that because we did not expect to measure the learning rate specific to sound in this phase, we actually used this phase as a motivation phase, giving free rewards to the animal to increase the number of port visits. The number of free reward was adjusted according to the motivation of the animal, so basically these measurements are biased by the procedure.

4) Page 3, line 194: the authors state that “learning speed depends more on the salience of the S- than of the S+ sound. The relative saliences...”. This is stated as fact, but is this actually a hypothesis? In the next line, the authors state “If X recruits less activity than Y...” from which it seems you have implicitly assumed that cortical recruitment = salience. But isn’t this what you want to show with the paper? Something about the logic is off here. The interchangeable use of “salience” and “cortical recruitment” at this point in the manuscript is not appropriate. On a related note, I would suggest that the authors do not use the S- < S+ terminology in Figure 3. Instead, could the authors use something easier to follow, such as “Sound A (S-), Sound C (S+)” for “S-<S+”.

We thank the referee for spotting this error, indeed the appropriate wording is “if X is less salient than Y, we expect learning to be slower when X is the S-.” as we now state in the manuscript. We also changed in figure 3b,c the terminology S- < S+. We however kept it in Fig. 3d as it is more compact and better summarizes the results at this point.

5) Page 12, line 255: the authors state “Most importantly here, the model has an explicit salience parameter, that corresponds to the activity of the S+ and S- populations”. Please clearly define which parameter you are referring to here – is it the 0.5 or 2x ratio of activities that you use? From the supplement, it appears that you have once again assumed cortical recruitment=salience, which might make the logic circular.

The referee is right that salience should not be mentioned here. We therefore changed the sentence into:

“Most importantly here, the activity level of the $\hat{S}+$ and $\hat{S}-$ populations, typically set to 1 (arbitrary units), can be varied in the model, allowing us to simulate the impact of cortical recruitment by setting either $\hat{S}-$ or $\hat{S}+$ activity level to 2.”

To avoid giving the impression of circular reasoning, we better introduced the purposes of the model at the beginning of the section where it is first described.

“In order to better understand why cortical recruitment specifically impacts the learning phase and why variability is larger when S+ recruits more activity than S-, we employed a recently developed model of the discrimination task.”

Last, we cleaned the confusion between salience and cortical recruitment in the text and in the supplementary note.

6) Fig. 5's legend makes a reference to learning "strategy", but its not clear why at this point. I would take "strategy" to mean a decision by the animal that following one set of heuristics works better than others for task performance. You could define this better in the text.

We changed the legend to "The effects of neuronal recruitment in the model are explained by the differential adjustment of synaptic weights during learning."

We also better defined strategy in the text.

"Specifically, when the \hat{S} - population recruits more activity, it is assigned to NoGo, while the \hat{C} population drives the Go response. In contrast, when the $\hat{S}+$ population recruits more activity, it is assigned to Go, and in this case, the \hat{C} population drives the NoGo response. Thus, different neuronal recruitment distributions result in different solutions of the binary discrimination problem based on the three available cues. These solutions could be seen as different strategies chosen by the model or eventually the animal during the learning process."

7) Page 20, line 443: This section to me is somewhat naive. There is a rich literature on different levels or patterns of recruitment for different sounds, and the authors' earlier study is one of them.

We rewrote this section.

"Several studies indicate that cortical recruitment can vary across stimuli, even when played at the same sound pressure level^{26,28,33}. These discrepancies may have multiple origins. First, it is well known that the mouse cochlea is more sensitive to some sound frequencies than others²⁹, which could explain the overrepresentations of sounds in the middle frequency range (10-30kHz) in cortex²⁸. In this case, cortical recruitment is expected to reflect recruitment throughout the auditory system, making it a good proxy for sound salience independent of whether the discrimination task requires auditory cortex⁴⁵⁻⁴⁷ or does not require it⁴⁸⁻⁵⁰. But a second source of recruitment differences may be the nonlinearities of cortical representations^{33,51-53}. For example, a recent study suggested that cortical response patterns can be invariant to changes in intensity⁵⁴. In this case, cortical recruitment should also depend on the higher level features composing the representation of a given sound and on how broadly these features are represented in cortex."

8) On a more philosophical note, consider the pup-retrieval behavior in mice. The pup distress calls, owing to their very high frequencies, likely recruit smaller extent of cortical activation than many other stimuli. Yet, pup distress calls are probably highly salient to dams (experienced moms). Wouldn't this be a counter-example to the claim that salient stimuli evoke more cortical activation? In general, I would like to see a brief discussion of alternative models.

This is a good remark. Pup calls trigger an array of interesting behaviors. However, these types of innate behaviors likely involve more hard-wired circuits that only partially overlap with the circuits involved in our Go/NoGo task.

The surprisingly large salience of pup calls in the context of maternal behavior could be due to strong and very specific connections between pup call representations and the centers triggering maternal responses, which do not exist for other sounds. In this case, it would be conceivable that the brain regions responsible for licking in our task, which presumably are not typically involved in the natural response to pup calls, are not more strongly connected to pup calls representations than to representations other sounds. If so, pup calls would not generate faster learning in dams performing our Go/NoGo task. In other words, we believe that the salience of a given stimulus can depend on

context and on the response it is meant to trigger. This could be modelled through differences in pre-existing connections.

In the manuscript, we actually show that learning speed not only depends on neuronal recruitment but also on initial connectivity patterns. Considering that “salience” is effectively the product of recruitment and connectivity (see our discussion pages 21-22, copied below), our model can in fact accommodate the counter-example of pup calls for maternal behavior, by supposing strong initial connectivity between pup calls representation and maternal responses.

“In particular, we had to introduce a more realistic multiplicative learning rule which renders learning speed not only dependent on neuronal recruitment (i.e. salience), but also on the current synaptic strength. This property has important consequences. First, it introduces variability in the relationship between recruitment and learning speed, through large inter-individual variations of the synaptic weights present at the beginning of the task. Second, the fact that learning speed is proportional to the product of neuronal recruitment and connectivity, allows the system more flexibility, in particular by compensating weak neuronal recruitment with stronger initial connections (see Supplemental Experimental Procedures). In our experiments, this phenomenon tends to stabilize learning speed, explaining why neuronal recruitment does not always impact the learning phase duration, except for particular initial conditions for which compensation occurs too slowly (Figs. 3-5). More generally, strong pre-established connections can help implementing fast learning for specific stimuli with innate meaning.”

We nevertheless discussed alternative models (also following a comment by reviewer 2) on page 21.

“For example, several studies indicate that attention can boost the activity of the neurons representing behaviorally relevant stimuli^{6,55,56} and thereby make it more discriminable from other stimuli⁵⁷. Also, several theoretical studies have proposed that attention impacts learning^{58,59} and some reinforcement learning models can account for such effects by dynamically weighting stimuli according to their predictive relevance⁶⁰. It will be an interesting research avenue to merge these alternative models of salience viewed as a top-down gain control effect, with models, as the one proposed here, focusing on the bottom-up neuronal implementation.”

Other concerns:

1) Zhaoping Li (2002) “A Saliency map in primary visual cortex” proposes a model that saliency monotonically correlates with V1 neuron firing rates and is mediated by layer 2/3 horizontal connections, and is a highly relevant reference that must be acknowledged.

We have now cited this reference in the introduction.

2) Figure 1A: I suggest moving the labels ‘A’, ‘B’ and ‘C’ to the top of the spectrograms, and the axis labels to the bottom. If the sounds are 70 ms long, then 0 and 70 should line up with the edges.

We have made these changes.

3) Page 6, line 133,135: percent “%” signs are incorrect.

Fixed.

4) Fig. 3b legend: individual learning curves are only shown for 4 mice, not 6.

We fixed this.

5) Fig. 3b,c: The x axis labels "1000 trials" are a bit confusing.

We have changed them, also in Fig. 5 and Fig. S2

6) In Fig. 5, could you mark, as in Fig. 3, the delay and learning phases?

We did it.

7) Line 676: Do you mean the U. Penn. viral vector core?

Yes, we added U. Penn.

Reviewer #2 (Remarks to the Author):

This paper, building on earlier work by Bathellier et al (2013), finds support for a multiplicative model of reinforcement learning in which stimulus "salience" (activation of the stimulus representation) influences learning speed after a delay period. The existence of a delay period, as pointed out in the earlier work, supports a multiplicative model in which learning dynamics depend on the synaptic weights. The authors find neurophysiological support for their model using an analysis of cortical recruitment and an elegant optogenetic manipulation.

Overall, I thought this was an interesting and well-written paper. The model is simple but makes non-trivial predictions that depart in important ways from earlier models.

We thank the referee for her/his thorough review of the manuscript. We have updated the manuscript according to the comments.

Major comments:

1/ It wasn't clear whether the decoding results shown in Fig. 2f were cross-validated (i.e., was the classifier evaluated on a held-out test set?).

Yes, the classifier was cross-validated. The training set contained 20 trials. The test set 10 trials. This information is now added to the Method section.

2/ The authors are worried about overfitting the model (p. 12), but I don't really understand why (of course, overfitting is an issue for all models, but I don't see anything special about this case). I think it would be useful for the authors to show that they can capture relationship between recruitment and learning shown in Fig 3, which relies on an analysis of individual differences in model parameter estimates.

We incorrectly described the issue as overfitting. The reason why it is not helpful to try and fit individual learning curves with individual differences in model parameters is because the fit is clearly underdetermined. We can easily fit individual learning curves already without having a recruitment difference for S+ and S-. This is because, we have two core parameters, the learning rate α and the learning rate asymmetry ν , which determine overall learning speed and learning speed difference between S- and S+. The recruitment parameters are redundant to these two parameters, so

we cannot unambiguously extract it from a fit involving all parameters. This is the reason why we chose to only check that, if we use parameters from a previous fit and change only the recruitment parameters, then we obtain the observed experimental effects.

We changed the related text in order to clarify this point.

“The dynamics of the model depend on the choice of its three core parameters (noise level, learning rate and asymmetry, see Methods) and of its initial synaptic weights. *As shown previously²⁵, individual learning curves could be fitted by adjusting these parameters, even without accounting for recruitment difference. Nevertheless, to test if the model captures the effect of recruitment, one can use a realistic set of parameters and test if asymmetric recruitment produces the effects seen during the behavior. We thus* looked at the qualitative behavior of the model using a set of parameters obtained in a previous group of experiments²⁵ by fitting the individual learning curves from 15 mice trained in a task identical to the one used in this study.”

2/ If the authors want to claim homogenous durations of the learning phase (p. 18), then this should be quantified in some way.

We now give the mean +/- SEM values and a statistical test (212 ± 117 trials for the large \hat{S} - vs 260 ± 220 for the small \hat{S} -, p = 0.26 Wilcoxon rank-sum test) page 19.

The authors have chosen to view "salience" in terms of stimulus activation. This is fine, but it might be worth discussing other views of stimulus salience in the perceptual and reinforcement learning literatures. For example, salience is sometimes conceptualized in terms of neuronal gain control. In the RL literature, there is a burgeoning discussion of attention (see for example work by Niv et al 2015) in which stimuli get weighted based on their predictive relevance. There is of course also a rich literature in animal learning (going back to classic works by Pearce and Hall 1980 and Mackintosh 1975) on how attention affects learning.

When discussing the model, we now refer to the idea that attention and associated neuronal gain control are also important for understanding salience.

“The idea that the amount of neuronal activity recruited by a stimulus influences behavior has been proposed in different contexts. For example, several studies indicate that attention can boost the activity of the neurons representing behaviorally relevant stimuli^{6,55,56} and thereby make it more discriminable from other stimuli⁵⁷. *Also, several theoretical studies have proposed that attention impacts learning^{58,59} and some reinforcement learning models can account for such effects by dynamically weighting stimuli according to their predictive relevance⁶⁰. It will be an interesting research avenue to analyze the relative contribution of bottom-up sound encoding and attentional top-down mechanisms to the level of cortical recruitment.*”

Minor comments:

typo, p. 4: ", ,"

fixed

p. 31, lines 779-781: missing math symbols

This problem occurred during PDF conversion at submission. We will hopefully fix that by submitting a PDF file instead of a Word file.

Fig 3b: caption says there are learning curves for 6 individual mice, but I only see 4. Relatedly, for 3c it says that there is n=6 for each curve, but then for 3d it says n=4.

Fixed: 4 mice for the example learning curves, then n=6 for each curves in 3c and 3d.

Fig 3d: computing a correlation with 4 data points is somewhat dubious.

We agree (see also referee 3). We added a data point and we removed this panel from the main figure (moved to Supplementary Fig. 2). The model actually does not predict a linear correlation between recruitment and learning phase duration (Fig. 4), so it is unnecessary to measure such correlation. We replaced the correlation analysis by a multivariate statistical test, previously shown in Supplementary Fig. 2, but more convincing than the correlation analysis. The Friedman test shows that over our 5 experiments the asymmetry of cortical recruitment is a significant factor contributing to learning phase duration variations. Here the associated change in the text:

*“We observed across the five sound pairs tested that learning phase duration was systematically longer when the S- sound recruited less activity than the S+ sound (**Fig. 3d**). A non-parametric analysis of variance showed this effect to be highly significant, while no systematic effect of cortical recruitment was observed for the delay phase (**Fig. 3d, Supplementary Fig. 2**).”*

Fig 3e: statistical quantification of the claims made about these results is needed.

We added this information: “mean normalized standard deviation difference: $93\% \pm 18\%$, n = 5 sound pairs, p = 0.008 Wilcoxon rank-sum test, Fig. 3e, Supplementary Fig. 2”

p. 19: "cortical recruitment affect" -> "cortical recruitment affects"

Fixed

typo, Supplement, p. 2: "the 2D the plane"

Done

Reviewer #3 (Remarks to the Author):

The authors present behavioral and physiological experiments that suggest that the number of cortical neurons that are activated in auditory cortex by a particular sound reflect the speed in which this sound can be learned as the no-go stimulus in an auditory task. The learning in this task tends to occur in two phases, a first phase where animals respond to the go and the no go stimulus and a second phase where the animal gradually improves by slowly learning to refrain from licking in presence of the no-go stimulus. The authors used the data from the number of neurons recruited to fit a previously published reinforcement learning model that reproduces this behavior, in that the recruitment is proportional to the speed of learning on the ramp phase and does not correlate with the duration of the initial phase. The authors then use an optogenetic training paradigm to show that

the recruitment is the crucial parameter and show that a neutral stimulus triggers licking depending on

its comparative size respect to the size of the go stimulus. If the neutral stimulus is smaller than the go stimulus, it will not trigger licking, but it would trigger licking if the neutral stimulus is larger than the go-stimulus. This effect can be explained by their reinforcement learning model.

However, there are serious concerns about the validity of the conclusions of this study given the extremely small sample size.

We thank the referee for his/her critical reading of the manuscript. Although we did not fully agree with some aspects of the major concerns raised by the referee (see our answers below), we have provided new data to address these concerns as quantitatively as possible.

Major concerns

1) The authors tested only 3 sounds and correlated the cortical recruitment with the behavioral saliency. Sounds can differ in many multiple dimensions and cortical recruitment might have correlated by chance with the relevant variable. With a sample of 3, it is hard to reach a conclusion. The authors should increase this number significantly and provide a justification for the number of sounds used.

It is indeed a limitation of the study, that due large inter-individual variability (which our model can account for), we needed to train at least 12 animals over at least 6 weeks to measure learning speed differences for a single pair of sounds. This is why we had produced only 4 data point for 4 sound pairs chosen among 5 sounds (and not just 3 sounds as suggested by the referee). We also agree, together with referee 2 (see minor comments), that measuring a correlation with only few points is probably too far-fetched.

However, our conclusions do not rely on the existence of a tight correlation between learning speed and cortical recruitment. We actually hypothesized that cortical recruitment influences learning speed such that learning speed will be slower when S- is less salient than S+ (as compared too S- > S+). This hypothesis is strongly supported by the data as we found that learning phase duration was significantly increased when recruitment for S- was larger than for S+ (4 sound pairs, 60 animals, assessed with a non-parametric ANOVA, the Friedman test, $p = 0.0032$, previous Suppl. Fig. 2). Our model reproduces this effect, but actually does not really predict a linear correlation between recruitment and learning speed (rather a non-linear relationship and high variability).

In order to further reinforce our statistical analysis, we have tested one further sound pair with a clear difference in cortical recruitment (see new Fig. 2) in additional behavioral experimental series. We have chosen two amplitude modulated white noise sounds at 1Hz vs 20Hz in order to explore a dimension of sound space (rhythmic AM modulations), different from the dimensions explored with the 4 other pairs whose differences rather rely on frequency content and direction of slow ramping intensity. We have removed the correlation analysis from the main figure, and now show in figure 3 the multivariate statistical analysis of Suppl. Fig. 2. With the 5 sound pairs (72 animals), the Friedman test gives a p-value of 0.0005. This indicates that, at a statistical level, the neuronal recruitment parameter has an impact on learning speed.

In the end, whatever the number of tested sound pairs, a statistical correlation cannot replace a causal link between two parameters. This is why we have designed experiments to causally relate the recruitment parameter with learning effects related to the salience parameter of our model.

2) The optogenetic manipulation recruitment is very different from the recruitment of cortical cells

by sounds. The sounds recruit interleaved population of neurons and optogenetics create topologically separated populations of neurons. This strong local stimulus might recruit inhibitory population of neurons in different way than sounds affecting the validity of the model. In addition, the more salient sound recruits at least 400% more neurons than the less salient stimuli, whereas as the more salient sound recruits 10% more neurons than the least salient sound. The authors should at least use optogenetic stimuli that better reflect the magnitude of recruitment differences of actual sounds.

The reviewer raises a valid point that the difference in cortical recruitment between pairs of sounds and pairs of optogenetic stimuli should be in a comparable range. Based on the size of the photo-stimulated areas, as we assume, the reviewer is concerned that recruitment differences are much larger for pairs optogenetic stimuli than for sound pairs.

In order to order to address this issue, we now measured the level of neuronal activity in a new set of experiments, using tetrode recordings through a slit in the cranial window, while shining the small or large disks of light (same size as during behavior) at different locations across the window (see Supplementary Fig. 4). We had provided in our former Figure 3 (now suppl. Fig. 2) the values of recruitment differences for each sound pair (noted as “ Δ recruitment” in Suppl. Fig. 2a). Δ recruitment is calculated as $2(S1-S2)/(S1+S2)$ where S1 and S2 are the recruitment value for each sound, and varied in the range of 30 to 90%. When applying the same calculation for the optogenetic stimuli, we obtain a value of 111 % which is in the range of strongest differences observed for sounds.*

*We furthermore calculated “ Δ recruitment” for many more sound pairs from a dataset of ~60,000 neurons and more than 100 clearly audible sounds (level \geq 60 dB, freq. range 4 to 32 kHz). The pairwise recruitment differences are ranging from 0 to 200% and ~20% of the sound pairs had recruitment differences above 111% (see below **Complementary data 1**). Together, this shows that the optogenetic stimuli, although they may not capture all aspects of natural sound responses in cortex, are well in the range of recruitment differences observed for sounds.*

Complementary data 1: Recruitment differences for 10500 sound pairs ordered by magnitude. Data set of ~60,000 neurons, recorded across the auditory cortex 7 mice using two-photon calcium imaging.

The main text was also updated to mention the electrophysiological measurements and evaluation of recruitment differences:

“Electrophysiological measurements of the population firing rate elicited by the small and large disks used in behavior showed that the recruitment difference between the stronger and weaker stimuli was

about 110 % (**Supplementary Fig. 4**), comparable with the population differences observed for sounds (**Fig. 4b**).

Reviewers' comments:

Reviewer #1 (Remarks to the Author):

The authors have addressed all my concerns, and in my view the manuscript is ready for publication.

Reviewer #2 (Remarks to the Author):

I am satisfied by the changes the authors have made in response to my comments.

Reviewer #3 (Remarks to the Author):

The authors have added some extra data. However the conclusions that cortical recruitment determines learning dynamics and strategy is not supported by the behavioral experiments using sounds given the low number of sounds used. In addition, the optogenetic manipulations do not reproduce the results from the sound experiments and make a different point. The manuscript needs better statistics to support their claims.

Reviewer #3 (Remarks to the Author):

The authors present behavioral and physiological experiments that suggest that the number of cortical neurons that are activated in auditory cortex by a particular sound reflect the speed in which this sound can be learned as the no-go stimulus in an auditory task. The learning in this task tends to occur in two phases, a first phase where animals respond to the go and the no go stimulus and a second phase where the animal gradually improves by slowly learning to refrain from licking in presence of the no-go stimulus. The authors used the data from the number of neurons recruited to fit a previously published reinforcement learning model that reproduces this behavior, in that the recruitment is proportional to the speed of learning on the ramp phase and does not correlate with the duration of the initial phase. The authors then use an optogenetic training paradigm to show that the recruitment is the crucial parameter and show that a neutral stimulus triggers licking depending on its comparative size respect to the size of the go stimulus. If the neutral stimulus is smaller than the go stimulus, it will not trigger licking, but it would trigger licking if the neutral stimulus is larger than the go-stimulus. This effect can be explained by their reinforcement learning model.

However, there are serious concerns about the validity of the conclusions of this study given the extremely small sample size.

We thank the referee for his/her critical reading of the manuscript. Although we did not fully agree with some aspects of the major concerns raised by the referee (see our answers below), we have provided new data to address these concerns as quantitatively as possible.

Major concerns

1) The authors tested only 3 sounds and correlated the cortical recruitment with the behavioral saliency. Sounds can differ in many multiple dimensions and cortical recruitment might have correlated by chance with the relevant variable. With a sample of 3, it is hard to reach a conclusion. The authors should increase this number significantly and provide a justification for the number of sounds used.

It is indeed a limitation of the study, that due large inter-individual variability (which our model can account for), we needed to train at least 12 animals over at least 6 weeks to measure learning speed differences for a single pair of sounds.

This is why we had produced only 4 data point for 4 sound pairs chosen among 5 sounds (and not just 3 sounds as suggested by the referee). We also agree, together with referee 2 (see minor comments),

that measuring a correlation with only few points is probably too far-fetched.

On page 4 the authors state " we chose three short, complex sounds (70ms duration) containing a large range of frequencies and temporal modulations, but normalized at equal mean pressure level (73dB SPL, Fig. 1a)". They characterize the recruitment of these 3 sounds as stated in page 5 : "We assessed recruitment of neural activity in the auditory cortex in response to these three sounds using two-photon calcium imaging in awake, passively listening mice, a technique that offers access to large samples of neuronal activity" .

However, our conclusions do not rely on the existence of a tight correlation between learning speed and cortical recruitment. We actually hypothesized that cortical recruitment influences learning speed such that learning speed will be slower when S- is less salient than S+ (as compared too S- > S+). This hypothesis is strongly supported by the data as we found that learning phase duration was significantly increased when recruitment for S- was larger than for S+ (4 sound pairs, 60 animals, assessed with a non-parametric ANOVA, the Friedman test, $p = 0.0032$, previous Suppl. Fig. 2). Our model reproduces this effect, but actually does not really predict a linear correlation between recruitment and learning speed (rather a non-linear relationship and high variability).

In order to further reinforce our statistical analysis, we have tested one further sound pair with a clear difference in cortical recruitment (see new Fig. 2) in additional behavioral experimental series. We have chosen two amplitude modulated white noise sounds at 1Hz vs 20Hz in order to explore a dimension of sound space (rhythmic AM modulations), different from the dimensions explored with the 4 other pairs whose differences rather rely on frequency content and direction of slow ramping intensity. We have removed the correlation analysis from the main figure, and now show in figure 3 the multivariate statistical analysis of Suppl. Fig. 2. With the 5 sound pairs (72 animals), the Friedman test gives a p-value of 0.0005. This indicates that, at a statistical level, the neuronal recruitment parameter has an impact on learning speed.

The p-value of 0.0005 indicates that animals indeed found that the easiness for withholding licking correlated with the degree of cortical recruitment for these particular 3 or 5 sounds. As pointed out by this and the other reviewers, the previous p-value does not permit us to make inferences about the effects of cortical recruitment. The relevant p-value can be calculated by determining how likely is it to find similar or stronger correlations (linear or not) in random variables? In order to illustrate these point I have provided with some cartoon simulations so illustrate the perils of inferring correlations from small number of measurements.

Lets be conservative and assume that the dimensionality of the sound space is just 200 dimensions. We will assume that these dimensions are random. I will also assume that the authors have measured large number of animals so they can determine these random variables with precision. From these random variables, I will try to find the correlation of learning easiness, which I will quantify as 1,2,or 3. Notice that these are random variables, so there should not be any correlation between them and the numbers 1,2, and 3.

As the following matlab code illustrates, when the number of tested sounds is so low as the ones used in this manuscript, most of these random variables will produce strong correlations. As shown by the simulation, the problem can mitigated by increasing the number of sounds used.

```
n_dimensions=200;  
n_sounds=3;
```

```
% Create a matrix of random measurements.  
mean_values=randn(n_dimensions,n_sounds);  
for k_d=1:n_dimensions  
x=corrcoef(1:n_sounds,mean_values(k_d,:));
```

```

corr_coef(k_d)=x(2,1);

end
figure, subplot(3,1,1) ,hist(corr_coef)
xlabel('Correlation coefficients')
ylabel('Number of dimensions')
title('3 sounds')
axis([-1 1 0 60])

n_sounds=5;

% Create a matrix of random measurements.
mean_values=randn(n_dimensions,n_sounds);
for k_d=1:n_dimensions
x=corrcoef(1:n_sounds,mean_values(k_d,:));
corr_coef(k_d)=x(2,1);

end
subplot(3,1,2) ,hist(corr_coef)
xlabel('Correlation coefficients')
ylabel('Number of dimensions')
title('5 sounds')
axis([-1 1 0 60])

n_sounds=20;
% Create a matrix of random measurements.
mean_values=randn(n_dimensions,n_sounds);
for k_d=1:n_dimensions
x=corrcoef(1:n_sounds,mean_values(k_d,:));
corr_coef(k_d)=x(2,1);
end
subplot(3,1,3) ,hist(corr_coef)
xlabel('Correlation coefficients')
ylabel('Number of dimensions')
title('20 sounds')
axis([-1 1 0 60])

```

In the end, whatever the number of tested sound pairs, a statistical correlation cannot replace a causal link between two parameters.

The reviewer disagrees with this statement. It is possible to be quantitative by being explicit about the number of dimensions of the auditory space and determine how many sounds are required to reach a valid conclusion between some physiology metric and behavioral relevance

This is why we have designed experiments to causally relate the recruitment parameter with learning effects related to the salience parameter of our model.

2) The optogenetic manipulation recruitment is very different from the recruitment of cortical cells by sounds. The sounds recruit interleaved population of neurons and optogenetics create topologically

separated populations of neurons. This strong local stimulus might recruit inhibitory population of neurons in different way than sounds affecting the validity of the model. In addition, the more salient sound recruits at least 400% more neurons than the less salient stimuli, whereas as the more salient sound recruits 10% more neurons than the least salient sound. The authors should at least use optogenetic stimuli that better reflect the magnitude of recruitment differences of actual sounds. The reviewer raises a valid point that the difference in cortical recruitment between pairs of sounds and pairs of optogenetic stimuli should be in a comparable range. Based on the size of the photostimulated areas, as we assume, the reviewer is concerned that recruitment differences are much larger for pairs optogenetic stimuli than for sound pairs.

In order to address this issue, we now measured the level of neuronal activity in a new set of experiments, using tetrode recordings through a slit in the cranial window, while shining the small or large disks of light (same size as during behavior) at different locations across the window (see Supplementary Fig. 4). We had provided in our former Figure 3 (now suppl. Fig. 2) the values of recruitment differences for each sound pair (noted as " Δ recruitment" in Suppl. Fig. 2a). Δ recruitment is calculated as $2*(S1-S2)/(S1+S2)$ where S1 and S2 are the recruitment value for each sound, and varied in the range of 30 to 90%. When applying the same calculation for the optogenetic stimuli, we obtain a value of 111 % which is in the range of strongest differences observed for sounds.

We furthermore calculated " Δ recruitment" for many more sound pairs from a dataset of $\sim 60,000$ neurons and more than 100 clearly audible sounds (level ≥ 60 dB, freq. range 4 to 32 kHz). The pairwise recruitment differences are ranging from 0 to 200% and $\sim 20\%$ of the sound pairs had recruitment differences above 111% (see below Complementary data 1). Together, this shows that the optogenetic stimuli, although they may not capture all aspects of natural sound responses in cortex, are well in the range of recruitment differences observed for sounds.

Complementary data 1:

Recruitment differences for 10500 sound pairs ordered by magnitude. Data set of $\sim 60,000$ neurons, recorded across the auditory cortex 7 mice using two-photon calcium imaging.

The main text was also updated to mention the electrophysiological measurements and evaluation of recruitment differences:

"Electrophysiological measurements of the population firing rate elicited by the small and large disks used in behavior showed that the recruitment difference between the stronger and weaker stimuli was about 110 % (Supplementary Fig. 4), comparable with the population differences observed for sounds (Fig. 4b).

The authors compare the increase in firing rate across all neurons and find 6 times less activity than the large disk that is 0.36 Hz versus 1.7 Hz. However, this is not the relevant parameter. The relevant parameter is the number of neurons that have significant responses to the light compared to the number of neurons that have significant responses to the sounds. For the sounds used, the number of responsive neurons fluctuated between 35% and 45%, that is, a difference of 10%. The authors do not provide this number for the optogenetic stimulation, so we will assume that optogenetic large stimulation circle would recruit 400% more neurons than the smaller circle. We can see the extreme difference between these two regimes by comparing figures 6d and 4 c. The auditory stimuli produced a large difference in learning rates and these differences disappear when using the optogenetic stimulation. The authors found a different effect that bears no relationship to their previous finding.

** See Nature Research's author and referees' website at www.nature.com/authors for information about policies, services and author benefits

The authors have added some extra data. However, the conclusions that cortical recruitment determines learning dynamics and strategy is not supported by the behavioral experiments using sounds given the low number of sounds used. In addition, the optogenetic manipulations do not reproduce the results from the sound experiments and make a different point. The manuscript needs better statistics to support their claims.

Reviewer #3 (Remarks to the Author):

The authors present behavioral and physiological experiments that suggest that the number of cortical neurons that are activated in auditory cortex by a particular sound reflect the speed in which this sound can be learned as the no-go stimulus in an auditory task. The learning in this task tends to occur in two phases, a first phase where animals respond to the go and the no go stimulus and a second phase where the animal gradually improves by slowly learning to refrain from licking in presence of the no-go stimulus. The authors used the data from the number of neurons recruited to fit a previously published reinforcement learning model that reproduces this behavior, in that the recruitment is proportional to the speed of learning on the ramp phase and does not correlate with the duration of the initial phase. The authors then use an optogenetic training paradigm to show that the recruitment is the crucial parameter and show that a neutral stimulus triggers licking depending on its comparative size respect to the size of the go stimulus. If the neutral stimulus is smaller than the go stimulus, it will not trigger licking, but it would trigger licking if the neutral stimulus is larger than the go-stimulus. This effect can be explained by their reinforcement learning model.

However, there are serious concerns about the validity of the conclusions of this study given the extremely small sample size.

We thank the referee for his/her critical reading of the manuscript. Although we did not fully agree with some aspects of the major concerns raised by the referee (see our answers below), we have provided new data to address these concerns as quantitatively as possible.

Major concerns

1) The authors tested only 3 sounds and correlated the cortical recruitment with the behavioral saliency. Sounds can differ in many multiple dimensions and cortical recruitment might have correlated by chance with the relevant variable. With a sample of 3, it is hard to reach a conclusion. The authors should increase this number significantly and provide a justification for the number of sounds used.

It is indeed a limitation of the study, that due large inter-individual variability (which our model can account for), we needed to train at least 12 animals over at least 6 weeks to measure learning speed differences for a single pair of sounds.

This is why we had produced only 4 data point for 4 sound pairs chosen among 5 sounds (and not just 3 sounds as suggested by the referee). We also agree, together with referee 2 (see minor comments), that measuring a correlation with only few points is probably too far-fetched.

On page 4 the authors state " we chose three short, complex sounds (70ms duration) containing a large range of frequencies and temporal modulations, but normalized at equal mean pressure level

(73dB SPL, Fig. 1a)". They characterize the recruitment of these 3 sounds as stated in page 5 : "We assessed recruitment of neural activity in the auditory cortex in response to these three sounds using two-photon calcium imaging in awake, passively listening mice, a technique that offers access to large samples of neuronal activity".

This quote is out of context. The referee forgets to mention that on page 6, we then report measurements of recruitment for 4 further sounds (Fig. 2g and h) and that our statistical assessment of the effect of recruitment on learning speed in Fig. 3d is based on 5 sound pairs and 7 sounds.

However, our conclusions do not rely on the existence of a tight correlation between learning speed and cortical recruitment. We actually hypothesized that cortical recruitment influences learning speed such that learning speed will be slower when S- is less salient than S+ (as compared too S- > S+). This hypothesis is strongly supported by the data as we found that learning phase duration was significantly increased when recruitment for S- was larger than for S+ (4 sound pairs, 60 animals, assessed with a non-parametric ANOVA, the Friedman test, $p = 0.0032$, previous Suppl. Fig. 2). Our model reproduces this effect, but actually does not really predict a linear correlation between recruitment and learning speed (rather a non-linear relationship and high variability).

In order to further reinforce our statistical analysis, we have tested one further sound pair with a clear difference in cortical recruitment (see new Fig. 2) in additional behavioral experimental series. We have chosen two amplitude modulated white noise sounds at 1Hz vs 20Hz in order to explore a dimension of sound space (rhythmic AM modulations), different from the dimensions explored with the 4 other pairs whose differences rather rely on frequency content and direction of slow ramping intensity. We have removed the correlation analysis from the main figure, and now show in figure 3 the multivariate statistical analysis of Suppl. Fig. 2. With the 5 sound pairs (72 animals), the Friedman test gives a p-value of 0.0005. This indicates that, at a statistical level, the neuronal recruitment parameter has an impact on learning speed.

The p-value of 0.0005 indicates that animals indeed found that the easiness for withholding licking correlated with the degree of cortical recruitment for these particular 3 or 5 sounds. As pointed out by this and the other reviewers, the previous p-value does not permit us to make inferences about the effects of cortical recruitment. The relevant p-value can be calculated by determining how likely is it to find similar or stronger correlations (linear or not) in random variables? In order to illustrate these point I have provided with some cartoon simulations so illustrate the perils of inferring correlations from small number of measurements.

Lets be conservative and assume that the dimensionality of the sound space is just 200 dimensions. We will assume that these dimensions are random. I will also assume that the authors have measured large number of animals so they can determine these random variables with precision. From these random variables, I will try to find the correlation of learning easiness, which I will quantify as 1,2, or 3. Notice that these are random variables, so there should not be any correlation between them and the numbers 1,2, and 3.

As the following matlab code illustrates, when the number of tested sounds is so low as the ones used in this manuscript, most of these random variables will produce strong correlations. As shown by the simulation, the problem can mitigated by increasing the number of sounds used.

```

n_dimensions=200;
n_sounds=3;

% Create a matrix of random measurements.
mean_values=randn(n_dimensions,n_sounds);
for k_d=1:n_dimensions
x=corrcoef(1:n_sounds,mean_values(k_d,:));
corr_coef(k_d)=x(2,1);

end
figure, subplot(3,1,1),hist(corr_coef)
xlabel('Correlation coefficients')
ylabel('Number of dimensions')
title('3 sounds')
axis([-1 1 0 60])

n_sounds=5;

% Create a matrix of random measurements.
mean_values=randn(n_dimensions,n_sounds);
for k_d=1:n_dimensions
x=corrcoef(1:n_sounds,mean_values(k_d,:));
corr_coef(k_d)=x(2,1);

end
subplot(3,1,2),hist(corr_coef)
xlabel('Correlation coefficients')
ylabel('Number of dimensions')
title('5 sounds')
axis([-1 1 0 60])

n_sounds=20;
% Create a matrix of random measurements.
mean_values=randn(n_dimensions,n_sounds);
for k_d=1:n_dimensions
x=corrcoef(1:n_sounds,mean_values(k_d,:));
corr_coef(k_d)=x(2,1);
end
subplot(3,1,3),hist(corr_coef)
xlabel('Correlation coefficients')
ylabel('Number of dimensions')
title('20 sounds')
axis([-1 1 0 60])

```

We thank the referee for developing his/her concern. We have run the simulations provided. They reproduce the well-known phenomenon, captured by the tests assessing significance of correlation values, that high correlations are more likely for a low sample number N . Thus, a high correlation has little meaning for low N . We would like to point out that in the referee's simulation, the so-called

'sound space dimension' parameter, set arbitrarily at 200, actually only corresponds to the number of repeats for the random generation of N=3, 5 or 20 values. If one increases the 'dimension' one gets a less noisy histogram of correlation values but the phenomenon remains the same. This can be easily seen by running the proposed simulation while setting the 'n_dimensions' parameter to lower values. So, the dimensionality issue should be taken out of the referee's concern.

We think that there is a misunderstanding about this point, because we had fully acknowledged in our previous review that correlation has little meaning for low N. As a consequence, we had removed the correlation analysis from the main text (the correlation plot is now in Supplementary Figure and the correlation is reported as not significant). To replace the correlation measure, we had statistically tested whether or not the sign of the difference between recruitment for the two discriminated sounds impacts learning speed. We had shown that it is the case with a p-value of 0.0005. Note that this multivariate analysis was actually present in the original submission to complement the correlation measure. This analysis obviously supports a conclusion that is different and less broad than the correlation analysis. The existence of a significant correlation (not the case for our data, see Suppl. Fig. 2) would allow us to conclude that the magnitude of recruitment difference matters for the magnitude of learning speed differences. Our multivariate analysis only allows us to conclude that the sign of the recruitment difference matters for the sign of the learning speed difference. However, as acknowledged by the other referees, this conclusion is sufficient to say that there is a statistical effect of recruitment on learning speed.

Furthermore, in order to test whether, in principle, 5 sounds are enough to detect an effect of the sign of recruitment difference on the sign of learning speed differences, we have run a simulation similar to the one proposed by the referee (see below). **It shows that the probability to observe by chance the same sign of learning speed differences for 5 sound pairs, as we observe when ordering these pairs based on cortical recruitment, is only about 3%** (this can also be calculated through binomial probability as $1/2^5$). In our case, this result is further reinforced by the fact that the sign differences are consistently observed across cohorts of several mice yielding the p-value of 0.0005.

```
n_repeats=2000;
n_sounds=5;
% Create a matrix of random measurements.
mean_values=randn(n_repeats,n_sounds);
for k_d=1:n_repeats
    a=sum(mean_values(k_d,:)>0);
    N_positive(k_d)=a;
end
figure, subplot(1,1,1), hist(N_positive,[0:n_sounds]);
xlabel('Number of observations with + sign')
ylabel('Number of repeats')
title('5 sounds')
```

To clarify these points, we made the following changes in the text:

- We removed the word correlate from the title of Fig. 3, which becomes "**Cortical recruitment differences impacts learning phase duration.**"

- We provided an assessment of the probability to obtain the observed result by randomly selecting unrelated sound properties:

“We observed across the five sound pairs tested that learning phase duration was systematically longer when the S- sound recruited less activity than the S+ sound (**Fig. 3d**), knowing that the probability to obtain by chance 5 similarly oriented effects (e.g. through random selection of other sound properties unrelated to neuronal recruitment) is only about 3% (2^{-5} , from binomial probabilities).”

In the end, whatever the number of tested sound pairs, a statistical correlation cannot replace a causal link between two parameters.

The reviewer disagrees with this statement. It is possible to be quantitative by being explicit about the number of dimensions of the auditory space and determine how many sounds are required to reach a valid conclusion between some physiology metric and behavioral relevance

We only meant that causal and correlative proofs are not considered of equal value by the scientific community as eventually a correlation can exist without direct causal link. Again, we had followed the referee’s previous comment and removed correlation analysis to replace it with a statistically sound analysis (see above).

The reviewer raises a valid point that the difference in cortical recruitment between pairs of sounds and pairs of optogenetic stimuli should be in a comparable range. Based on the size of the photostimulated areas, as we assume, the reviewer is concerned that recruitment differences are much larger for pairs optogenetic stimuli than for sound pairs.

In order to address this issue, we now measured the level of neuronal activity in a new set of experiments, using tetrode recordings through a slit in the cranial window, while shining the small or large disks of light (same size as during behavior) at different locations across the window (see Supplementary Fig. 4). We had provided in our former Figure 3 (now suppl. Fig. 2) the values of recruitment differences for each sound pair (noted as “ Δ recruitment” in Suppl. Fig. 2a). Δ recruitment is calculated as $2 \cdot (S1 - S2) / (S1 + S2)$ where S1 and S2 are the recruitment value for each sound, and varied in the range of 30 to 90%. When applying the same calculation for the optogenetic stimuli, we obtain a value of 111 % which is in the range of strongest differences observed for sounds.

We furthermore calculated “ Δ recruitment” for many more sound pairs from a dataset of ~60,000 neurons and more than 100 clearly audible sounds (level \geq 60 dB, freq. range 4 to 32 kHz). The pairwise recruitment differences are ranging from 0 to 200% and ~20% of the sound pairs had recruitment differences above 111% (see below Complementary data 1). Together, this shows that the optogenetic stimuli, although they may not capture all aspects of natural sound responses in cortex, are well in the range of recruitment differences observed for sounds.

The main text was also updated to mention the electrophysiological measurements and evaluation of recruitment differences:

“Electrophysiological measurements of the population firing rate elicited by the small and large disks used in behavior showed that the recruitment difference between the stronger and weaker stimuli was about 110 % (Supplementary Fig. 4), comparable with the population differences observed for sounds (Fig. 4b).

The authors compare the increase in firing rate across all neurons and find 6 times less activity than the large disk that is 0.36 Hz versus 1.7 Hz. However, this is not the relevant parameter. The relevant parameter is the number of neurons that have significant responses to the light compared to the

number of neurons that have significant responses to the sounds. For the sounds used, the number of responsive neurons fluctuated between 35% and 45%, that is, a difference of 10%. The authors do not provide this number for the optogenetic stimulation, so we will assume that optogenetic large stimulation circle would recruit 400% more neurons than the smaller circle.

The referee confirms that we have convincingly shown in our revised manuscript that “mean population firing rates” in AC for optogenetics and for real sound are in similar ranges. However, he/she argues that the “mean population firing rate” is not the relevant parameter to measure recruitment, but rather the “fraction of responsive neurons”.

We had mentioned in the introduction that classical, experimentally verified synaptic learning rules such as the BCM rule (Bienenstock, Cooper, Munro, rate based rule) or modern spike-based STDP learning rules (see ref 10-14 in the manuscript) generate synaptic updates that vary together with the number of action potentials fired by each neuron, and not together with an all-or-none “activation variable” which does not account for the number of spikes. This theoretical point, which is one of the foundations of our study imposes to use “mean population firing rate” as the measure of cortical recruitment. **Based on these arguments, we disagree with the referee’s statement that mean population firing rate “is not the relevant parameter”.**

We added the following sentence in the result section to clarify this point: “Because typical learning rules are dependent on pre-synaptic firing rate¹⁰⁻¹⁴, we first measured the amplitude of the mean-deconvolved calcium signals, a proxy for neuronal firing rate³²,”

Moreover, the “fraction of responsive neurons” suffers from the limitation that it strongly depends on its associated statistical threshold which must be somewhat arbitrarily chosen. For the sounds labelled as A and C in Fig. 2e, we measured that 35% and 45% of all neurons responded based on a one-sided Sign test and an alpha-value (statistical threshold) of 5% (note that the one-sided test was used here by mistake and the figure was updated to include results from a two-sided test). As we now use a two-sided test, the fractions of responsive neurons become 18% and 25% respectively with alpha=5% but change to 11.5% and 7% with alpha=1%. The three pairs of values above yields fairly different “ Δ recruitment” values (25%, 32% and 49%), calculated as $2*(S1-S2)/(S1+S2)$. **Hence, the “fraction of responsive neurons” is not a robust way to estimate neuronal recruitment for sounds**, as we now mention in the results:

“But note that the fraction of responsive neurons strongly depends on statistical threshold.”

Nevertheless, to further confirm that our optogenetic calibration experiments do not rely on “out-of-range” stimuli, we decided to re-estimate as precisely as possible the “ Δ recruitment” for optogenetic stimuli based on both mean firing rate and fraction of responsive cells. In particular, we used a more stringent estimate of the spatial integral of the response to light disks across AC, which now accounts for the fact that the sampling grid of small disk centers spans a larger space than the grid of large disks centers (see details added in the Methods). Updated “ Δ recruitment” index for population firing rate are 69% (Suppl. Fig. 4c,d). Moreover, using a double-sided Signed test and an alpha value of 5%, the small and big disk approximately recruit 10% and 25% of the neurons across the area of the sampling grid. This translates into a “ Δ recruitment” value of $2*(S1-S2)/(S1+S2) = (25-10)/(25+10+10+10) = 55\%$. All these values are in the same order of magnitude as the values observed for sounds.

These values are now reported in the manuscript in the legend of Suppl. Fig. 4.

We can see the extreme difference between these two regimes by comparing figures 6d and 4 c. The auditory stimuli produced a large difference in learning rates and these differences disappear when using the optogenetic stimulation. The authors found a different effect that bears no relationship to their previous finding.

We have already proposed in the Result part a different explanation about the fact that long learning phase durations are not observed for the optogenetic experiments.

“Importantly, in this task setting, the stringent definition of the common \hat{C} population, activated during the initial motivation training, likely resulted in the systematic establishment of strong initial connections for this population at the beginning of the discrimination training, leading to homogenous durations of the learning phase (212 ± 117 trials for the large \hat{S} - vs 260 ± 220 for the small \hat{S} -, $p = 0.26$, Wilcoxon rank-sum test, see also **Fig. 6d**) independent of recruitment (**Figs. 4-5**).”

Note also that many animals trained to sounds showed fast learning rates independent of S-recruitment as shown in the distribution plots of Fig. 3e (note that Fig. 4c mentioned by the referee represents results from the model not sound-based experiments). It is thus not abnormal that fast learning rates are observed in the Fig. 6d for the optogenetic experiment. Remarkably, our model reproduces this effect without any fitting procedure, and provides a more robust prediction about the role of cortical recruitment which we decided to test using a causal strategy.

Reviewers' Comments:

Reviewer #3:

Remarks to the Author:

This referee's concerns have not been addressed and the conclusion stated in the title that "Cortical recruitment determines learning dynamics and strategy" is weakly supported by the data. No new data nor analysis is being presented. As it stands, the manuscript is not suitable for publication.

Major concerns

1) The authors seem to be a conceptual mistake in reporting the effect of cortical recruitment on learning speed. The Friedman test, ($p = 0.0005$) tested that indeed, for the 5 sounds that they used, the ones with largest recruitment did produce the fastest learning. This p-value refers to only these 5 sounds and no inference could be made to any other sounds. On the other hand, the p-value of 0.03 is pertinent to how the measurement chosen, cortical recruitment, determines the learning dynamics and strategy, i.e. how we could generalize from these results. Although a p value of 0.03 is considered marginally significant, it is not corrected for multiple comparisons, as it was found not to correlate with learning delay but with learning speed.

The authors need to increase the number of sound pairs used before we can conclude that cortical recruitment determines learning dynamics.

2) The authors claim that the actual number of neurons recruited by the optogenetical stimulus is not relevant, and prefer to report the average activation across the population as the relevant parameter. However, this is a problematic assumption. By having a stimulus that directly activates an area that is 4 times larger as the S- is well outside the range of differences between sounds. This is very relevant for the interpretation of the results. For instance, it is way more plausible that the reason that the C stimulus by itself is perceived as a go stimulus when the S- is a large disc is that the large and synchronous stimulation triggers a massive recurrent cortical inhibition that completely masks the C stimulus when paired with S-. Therefore the animal will perceive C as part of the S+ and reacts that way. With the data presented in the manuscript so far, this very mechanistic explanation is more plausible than the reinforcement learning presented by the authors. The authors need to show that the large disc stimulation does not triggers large waves of inhibition that will mask the activity of the C stimulus. The use of extracellular recordings make it necessary to have large levels of spontaneous activity, in order to determine the presence of inhibition.

3) The authors claim that optogenetic stimulation works in a similar regimen and that the lack of differences in the learning might be related to the differences in the intersection C population for auditory stimuli versus optogenetic stimuli. They state that:

"in this task setting, the stringent definition of the common C population, activated during the initial motivation training, likely resulted in the systematic establishment of strong initial connections for this population at the beginning of the discrimination training, leading to homogenous durations of the learning phase (212 ± 117 trials for the large S- vs 260 ± 220 for the small S-, $p = 0.26$, Wilcoxon rank-sum test, see also Fig. 6d) independent of recruitment (Figs. 4-5)."

The common C population does exist in the sound case, and in fact, it can be easily calculated as the population of cells that are activated by the S+ and S- sounds. How many neurons were activated by both sounds? How this is different from the optogenetic case? The authors should use their imaging data to fit their model, and not rely only on the behavioral data, as this makes the logic circular.

Minor concerns

Could the authors include all their data in their figures?

On Figure 2 F there is only data from the original 3 sounds. The same applies for the panels in figure 3, they should include the data on the extra sounds. When the authors include more sounds in their database, they should also determine that they have similar detectabilities.

1) The authors seem to be a conceptual mistake in reporting the effect of cortical recruitment on learning speed. The Friedman test, ($p = 0.0005$) tested that indeed, for the 5 sounds that they used, the ones with largest recruitment did produce the fastest learning. This p-value refers to only these 5 sounds and no inference could be made to any other sounds. On the other hand, the p-value of 0.03 is pertinent to how the measurement chosen, cortical recruitment, determines the learning dynamics and strategy, i.e. how we could generalize from these results. Although a p value of 0.03 is considered marginally significant, it is not corrected for multiple comparisons, as it was found not to correlate with learning delay but with learning speed.

The authors need to increase the number of sound pairs used before we can conclude that cortical recruitment determines learning dynamics.

There is no conceptual mistake. Our data indeed show very strongly that for the five sounds used, the ones with largest recruitment did produce the fastest learning based on statistical testing. Thus across five experiments there is, in the statistical sense, an effect of recruitment on learning speed.

The referee raised the concern that these results could not be generalized to other experiments made with other sounds. No study could ever demonstrate that one result observed within a set of defined conditions (here the set of sounds chosen) will be observed in any other set of different conditions (for any set of sounds). So, the misconception is on the referee's side and it is not scientifically reasonable to ask for experimentally demonstrating that our results are generalizable. No statistical approach exists for this purpose.

Evaluation of how generalizable scientific results are, may be done in two ways. 1/ Reproducing the experiment in various conditions (as suggested by the referee). 2/ And even better, proposing a model that gives a general mechanism by which the results occur. The model can be tested in different ways to evaluate the scope of its validity (and hence how generalizable it is). Yet, there exists no statistical criterion for the generality of a model. Statistics is only useful to test if the model matches measurements made in a particular condition.

In this manuscript, we followed both approaches. We have repeated 5 times the measurements of learning speed and found with excellent statistical control that results match the qualitative model stating that higher recruitment tend to yield higher learning speed. We have then compared these results to a quantitative model WITHOUT ANY FITTING OF THIS MODEL TO THE DATA and showed that our five experiments match the model when considering the sign of mean learning speed and learning speed variance differences between low and high recruitment. Our model also matches the experiments in predicting that there should be no influence of recruitment on learning delays.

Very concretely, in his/her previous comment, the referee has asked us to evaluate a different model stating that some random undefined sound parameters (not controlled in our experiments) could explain by chance the effects seen in our measurements. Because his/her model is not precisely specified, the best that can be done is to calculate the chance that a random influence would yield 5 out of 5 times an effect of the same sign. This probability is 0.03. Thus the referee's model has little chance to match the results that we obtained for mean learning speed (5 out of 5 measurements show the same impact of recruitment) and for learning speed variance (again 5 out of 5 consistent measurements). Because the referee's model predicts no relation between mean and variance in our measurements, the probability that it matches the overall data by chance is $0.03^2 = 0.0009$. Based on this we can reject the referee's model against our model suggesting that "cortical recruitment determines learning dynamics". Moreover, the fact that our model and the referee's are equally good in capturing the absence of relation between recruitment and learning delays does not weaken the validity of our model against the referee's.

Yet, in order to clearly acknowledge the intrinsic limitations of having repeated the experiment in 72 mice but only 5 sound pairs, we inserted the following text in the Results:

” Together, these results obtained over a total of 72 mice, indicated clear relations between cortical recruitment and learning phase duration for the five pairs of sounds tested. It cannot be ruled out a priori that other, non-measured parameters of the sound representations could explain this dependency. Yet, in the hypothesis that these parameters would be randomly assigned to the tested sounds, the probability to obtain by chance a consistent result across five independent experiments would be only about 3% (2^{-5}), so we expect this eventuality to be rather unlikely.

A reinforcement learning model captures the effects of cortical recruitment on learning dynamics

In order to theoretically evaluate generality of these results, we wondered whether existing reinforcement learning models for this task actually predict these results in details, independent of sound quality. “

2) The authors claim that the actual number of neurons recruited by the optogenetical stimulus is not relevant, and prefer to report the average activation across the population as the relevant parameter. However, this is a problematic assumption. By having a stimulus that directly activates an area that is 4 times larger as the S- is well outside the range of differences between sounds. This is very relevant for the interpretation of the results. For instance, it is way more plausible that the reason that the C stimulus by itself is perceived as a go stimulus when the S- is a large disc is that the large and synchronous stimulation triggers a massive recurrent cortical inhibition that completely masks the C stimulus when paired with S-. Therefore the animal will perceive C as part of the S+ and reacts that way. With the data presented in the manuscript so far, this very mechanistic explanation is more plausible than the reinforcement learning presented by the authors. The authors need to show that the large disc stimulation does not triggers large waves of inhibition that will mask the activity of the C stimulus. The use of extracellular recordings make it necessary to have large levels of spontaneous activity, in order to determine the presence of inhibition.

The referee unfortunately overlooked in our answers to his/her previous comments that we quantified the number of neurons recruited by the optogenetic and auditory stimuli showing that they are comparable.

We do not see large waves of inhibition produced by the large optogenetic stimulus at the location of the C stimulus. We quantified the response produced by the large optogenetic stimulus 1 mm away from its center (location of the C stimulus). The results are now shown in the new Suppl. Fig. 4f, and indicate a slight increase of neuronal activity (instead of inhibition) above baseline.

The result of this analysis is now mentioned in the last paragraph of the Results.

“Note that these effects are unlikely to be caused by inhibition of \hat{C} when paired with the large $\hat{S}+$ or $\hat{S}-$ spot, as no inhibition from the larger spot was observed at the location of \hat{C} in calibration experiments (Supplementary Fig. 4).”

Note also that we added behavioral responses to the S+ and S- spots alone in Figure 6, which further confirm predictions of the model, even when counterintuitive, e.g. when the small C population drives the Lick response, the small S+ population alone does not drive it.

3) The authors claim that optogenetic stimulation works in a similar regimen and that the lack of differences in the learning might be related to the differences in the intersection C population for auditory stimuli versus optogenetic stimuli. They state that:

“in this task setting, the stringent definition of the common \hat{C} population, activated during the initial motivation training, likely resulted in the systematic establishment of strong initial connections for this population at the beginning of the discrimination training, leading to homogenous durations of the learning phase (212 ± 117 trials for the large \hat{S} - vs 260 ± 220 for the small \hat{S} -, $p = 0.26$, Wilcoxon rank-sum test, see also Fig. 6d) independent of recruitment (Figs. 4-5).”

The common C population does exist in the sound case, and in fact, it can be easily calculated as the population of cells that are activated by the S+ and S- sounds. How many neurons were activated by both sounds? How this is different from the optogenetic case? The authors should use their imaging data to fit their model, and not rely only on the behavioral data, as this makes the logic circular.

The referee misinterpreted our statements and ask for comparisons that are not justified.

The statement quoted by the referee only means that the difference between optogenetic and sounds, in our mind, is related to “the systematic establishment of strong initial connections” between the C and “Go” populations. It does not mean anything about a difference in the structure of C in sound and optogenetic experiments. We have however stated on page 17 that the C population likely extends beyond auditory neurons in the sound experiments “the neurons encoding common information between S+ and S- trials (\hat{C} population) likely code for multiple cues, including (i) the overlap of S+ and S- representations and (ii) all cues related to the decision to visit the lick port, and cannot be isolated”. We thus do not see how we could, in this case, simply look at the “overlap of S+ and S- representations” to disprove our statement about the differences between the “common” C populations in sound and optogenetics.

Minor concerns

Could the authors include all their data in their figures?

On Figure 2 F there is only data from the original 3 sounds. The same applies for the panels in figure 3, they should include the data on the extra sounds. When the authors include more sounds in their database, they should also determine that they have similar detectabilities.

This concern is not justified. We did include all our data in our figures. Figure 2 has imaging data for 7 sounds. Figure 3 has the data for the 5 sound pairs (out of 7 sound tested). And even all the data distributions are displayed in Supplementary figures.

Reviewers' Comments:

Reviewer #1:

Remarks to the Author:

N/A

Remarks to the Editor:

This is a difficult dispute to mediate because it appears that both the reviewer and the authors, over the previous review rounds, seem to have made their positions quite inflexible. But I think this deadlock is solvable - on the one hand, the reviewer could take into account the fact that these experiments are difficult to do and are time consuming. Some of the requests are thus somewhat impractical. The authors for their part could provide some additional analyses that do not require new experiments. I'm providing detailed comments below, but in my opinion, I broadly support the author's position on points 1 and 2, and the reviewer's position on point 3.

1) Reg. the concern that only a few sound-pairs are used

For this question, I think that the authors are reasonable in pointing out that it is impossible to prove that results will generalize to ALL sounds. How many sound pairs will satisfy the reviewer that the results are valid? Alternatively, what type of sound pair does the reviewer expect will prove to be the litmus test of the model? Since the reviewer has not mentioned specific pairs or numbers in his/her reviews, the authors rightfully do not want to spend months training mice only to realize that the reviewer will not be satisfied.

The authors have shown the relationship between recruitment and learning for three complex sound pairs (sounds A, B, and C), a noise-AM pair, and have used previous data for a ramped-damped pair and have shown consistent results. To me this is a reasonable 'variety' of sounds to have tried.

Part of the issue here is that the authors are quite brief in their discussion. I think a more thoughtful discussion of alternate models and the reviewer's concerns could settle this question. This could include possible examples of stimulus-pairs that could be counter examples to their findings. For example, in the first round of review, I had mentioned pup calls, which might excite few cortical neurons but could be of high salience. This may or may not be cortically mediated, but would be an interesting counterpoint to consider.

2) Reg. the number of neurons recruited

This was a valid point from the reviewer, but I think that in new Supp. Fig. 4f, the authors have provided recordings to show that they do not observe large waves of inhibition. I am satisfied with this explanation.

The reviewer does bring up the point of whether overall population activity or fraction of responsive cells is the appropriate metric for recruitment. While I understand the author's point about the effect of thresholding, I would suggest that the authors additionally provide the % active neurons in response to the AM noise pairs and the ramp pairs. This would help nail down exactly what the authors mean by stimuli that "produce more activity", and would assuage the reviewer as well.

3) Reg. the equivalence of sound and optogenetic stimulation

In this point, I think it is fair of the reviewer to ask if the authors can quantify the sizes of the C, S+ and S- from their imaging data. The authors have not directly addressed this.

From the available numbers, the fraction of active cells for sounds A, B, and C are 18%, 24% and 25% (page 6, line 9). For the optogenetic stimulation, they report small disk and large disk activate 10% and 42% of neurons (Supplement Page 4). Thus it does seem that the optogenetics behavior is being driven by "neural" differences that are substantially larger. Analyzing the neural data to determine what fraction of neurons commonly respond to sounds A, B, C, and estimating the relative proportion of neurons driven by C+small and C+large would provide a more appropriate comparison.

The reviewer is justified in asking for a more data-driven approach to the model. Currently, sounds A, B, C show an unknown but likely small difference between S+ and S- recruitment in terms of neuron numbers. The optogenetics use a 4-fold area difference. The model uses a 2-fold difference. While there are of course questions as to whether the percent of neurons or population activity is the right "population recruitment" metric to use, it would be nice for the authors to build a model be based on the fraction that is observed in the imaging data. The authors could present an additional supplemental figure showing this.

Point-by-point response to editorial concerns:

1) Reg. the concern that only a few sound-pairs are used

Editor: Please include a more thorough discussion of potential alternate models, and the Reviewer concerns. One suggestion is to include potential counter examples, such as the pup-calls as mentioned by Reviewer 1 in the earlier round of review.

It is important to discuss how generalizable our results may be given the small amount of sounds tested. Given that we still chose sounds with various properties, with which mice have no special experience, it is reasonable to think that, for such kind of “neutral” complex sounds, our results would hold. However, the referee is right that there may be counter-examples, particularly with sounds that have a peculiar behavioral meaning. Therefore, we have added a more thorough discussion of this issue, included pertinent pup-calls example, in the Discussion part.

“In this study, we have shown an effect of cortical recruitment level on learning speed significant for five different pairs of sounds with no particular meaning to the animal. This is not excluded that some sounds, in particular, sounds with learnt or innate meaning would show a different relationship. For example, pup calls are extremely salient to mothers but trigger little activity in cortex⁶⁴. This could be due to strong pre-existing wiring between cortical neurons responding to pup calls compensating for limited recruitment, at least for some behavioral situations. So, even if, as we show cortical recruitment plays a role in the salience of a sound, general theories of salience should also account for potential a priori meaning of stimulus, via pre-existing connections, as simple extensions of our model would suggest, or via more complex cognitive processes assigning value to the sounds.”

2) Reg. the number of neurons recruited

Editor: Please additionally provide the % active neurons in response to AM noise pairs and the ramp pairs as suggested by Reviewer 2 so that this metric can also be compared.

We have provided the fraction of responsive cells as insets in Fig. 2g&h, which both show a significant difference.

Edited caption: “The inset show the significantly different (χ^2 test, $p = 10^{-68}$) fraction of responsive cells (75% and 61% ; two-sided Signed test, $p < 0.05$).”

“The inset show the significantly different (χ^2 test, $p = 10^{-189}$) fraction of responsive cells (37% and 29% ; two-sided Signed test, $p < 0.05$).”

3) Reg. the equivalence of sound and optogenetic stimulation

Because the small and large disk activate 10% and 42% of the recording neurons in our calibration experiments, it seems that, during stimulation with both disks, we are recruiting more neurons than with sounds (activating ~20% of imaged neurons). However, the 42% value reported by the referee was an unfortunate typo (we actually copy pasted by mistake (including Hz units!) the 42.3 ± 5.7 Hz corresponding to firing rate values. The actual value is 25 ± 3 %.

It should be reminded that the fractions reported for the disks are obtained with a small array of electrode on a single location of auditory cortex while imaging is performed across the entire auditory cortex. So the values above are not absolutely comparable. It is indeed more relevant to measure the fraction of neurons commonly activated by sounds and optogenetic stimuli, as proposed by the referee.

Editor: Please analyze the neural data to determine what fraction of neurons commonly respond to sounds A, B, C, and estimate the relative proportion of neurons driven by C+small and C+large for a more appropriate comparison.

We have performed this analysis and provide it now in the result section describing our optogenetic experiments: “Electrophysiological measurements of the population firing rate elicited by the small and large spots showed that the recruitment difference between the stronger and weaker stimuli was about 69 % (Supplementary Fig. 4), comparable with the population recruitment differences observed for sounds (Fig. 4b; Supplementary Fig. 4). Also, we measured that the large disk activates about 2,5 more neurons than the small disk (Supplementary Fig. 5). Given that the \hat{C} spot

has a small diameter, we could evaluate the fraction of cells commonly activated by the S+ and S- stimuli within the cells activated by S+ and S- as $1 / (1+1+2.5) \approx 22\%$, similar to the fraction of cells commonly activated by sounds (e.g. 18%, 18%, 23% for sound pairs A-B, A-C, and B-C, s.e.m = 0.3%, binomial distribution). Thus the artificial stimuli, although not identical to sound responses had comparable characteristics.”

Editor: Please ensure that the model parameters are informed by the experimental data.

It is indeed an important question whether the recruitment parameter used in our model are similar to experiments. We had unfortunately not mentioned that point clearly enough in the main text. The reason why this escaped our attention is because we have at several places shown that the conclusions of the model do not qualitatively depend on the value of this parameter. This is shown in Figure 4b in which we have plotted the learning speed effect against recruitment in the model and in the data and shown that the same effect (yet with different magnitudes) is seen across a large range of recruitment difference values. This is also shown in Supplementary Figure 3 in which we had systematically varied recruitment (y-axis of all graphs). Finally, the effect seen with the optogenetic is predicted for any value of the recruitment parameter by our model, as demonstrated analytically (thus in an absolutely general manner, regarding the model of course). We have clarified this in the text in addition to providing the requested, self-speaking supplementary figure.

Editor: A more data-driven approach to the model is needed as follows. Although the appropriate population recruitment metric may be percent active neurons or population activity, please use the fraction of neurons observed in the imaging data to build the model and provide it as an additional Supplementary figure.

We have reproduced integrally Figure 5, which was done with a recruitment ratio of 2 (firing rate) with the recruitment ratio corresponding to fractions of recruited cells for sounds A and C ($1.4 = 25/18$). This is now displayed as Supplementary Figure 4 and shows no qualitative difference to figure 5 (but of course small quantitative differences). As we had already systematically explored the influence of the recruitment ratio in several figures (Fig. 4b, Supplementary Figure 3) and also analytically (showing that our conclusions do not qualitatively depend on the value of this ratio) we have quoted these elements together with the new figure.

Text parts in which we have quoted this supplementary figure:

“Most importantly here, the activity level of the \hat{S}^+ and \hat{S}^- populations, typically set to 1 (arbitrary units), can be varied in the model, allowing us to simulate the impact of cortical recruitment **for example** by setting either \hat{S}^- or \hat{S}^+ activity level to 2 (but note that this value can be widely varied without changing our conclusions, **Fig. 4b, Supplementary Fig. 3&4, Supplementary Experimental Procedures**).”

“Interestingly also, this provides a very general and testable prediction of reinforcement learning models using population activity as a salience parameter (**independent of the magnitude of recruitment differences**, see analytical arguments in Supplemental Experimental Procedures). The test would be to isolate and drive the neurons potentially corresponding to \hat{C} in the brain. Activation of the \hat{C} population alone should drive licking when S- recruits more activity, and should not drive licking when S- recruits less activity (**Fig. 5, Supplementary Figure 4**).”